

# A cloud identification algorithm over the Arctic for use with AATSR/SLSTR measurements

Soheila Jafariserajehlou[1], Linlu Mei[1], Marco Vountas[1], Vladimir Rozanov[1], John Philip Burrows[1], Rainer Hollmann[2]

[1]Institute of Environmental Physics, University of Bremen, Otto-Hahn-Allee 1, Bremen, 28359, Germany
[2]DWD – Deutscher Wetterdienst, Frankfurter Straße 135, 63067 Offenbach, Germany

*Correspondence to:* Soheila Jafariserajehlou (jafari@iup.physik.uni-bremen.de)

**Abstract.** The accurate identification of the presence of cloud in the ground scenes observed by remote sensing satellites is an end in itself. Our lack of knowledge of cloud at high latitudes increases the error and uncertainty in the evaluation and assessment of the changing impact of aerosol and cloud in a warming climate. A prerequisite for the accurate retrieval of Aerosol Optical Thickness, AOT, is the knowledge of the presence of cloud in a ground scene.

In this study observations of the up welling radiance in the visible (VIS), near infrared (NIR), shortwave infrared (SWIR), and the thermal infrared (TIR), coupled with solar extraterrestrial irradiance are used to determine the reflectance. We have developed a new cloud identification algorithm for application to the reflectance observations of Advanced Along-Track Scanning Radiometer (AATSR) on European Space Agency (ESA)-Envisat and Sea and Land Surface Temperature Radiometer (SLSTR) onboard the ESA Copernicus Sentinel-3A and -3B. The resultant AATSR/SLSTR Cloud Identification Algorithm (ASCIA) developed addresses the requirements for the study AOT at high latitudes and utilizes time-series measurements. It is assumed that cloud free surfaces have unchanged or little changed patterns for a given sampling period, whereas cloudy or partly cloudy scenes show much higher variability in space and time. In this method, the Pearson Correlation Coefficient (PCC) parameter is used to measure the 'stability' of the atmosphere-surface system observed by satellites. The cloud free surface is classified by analyzing the PCC values at the block scale $25\times25$ km$^2$. Subsequently, the reflection of 3.7 μm is used for accurate cloud identification at the scene level either $1\times1$ km$^2$ or $0.5\times0.5$ km$^2$. The ASCIA data product has been validated by comparison with independent observations e,g. Surface synoptic observations (SYNOP), AErosol RObotic NETwork (AERONET) and the following satellite-products from i) ESA standard cloud product from AATSR L2 nadir cloud flag, ii) one method based on clear-snow spectral shape developed at IUP Bremen (Istomina et al., 2010), which we call, ISTO, iii) Moderate Resolution Imaging Spectroradiometer (MODIS). In comparison to ground based SYNOP measurements, we achieved a promising agreement better than 95 % and 83 % within ±2 and ±1 okta respectively. In general, ASCIA shows an improved performance in comparison to other algorithms applied to AATSR measurements for cloud identification at high latitudes.

## 1 Introduction

The large trends in warming over the Arctic in recent decades, has received much attention from the global and regional climate change research community (Wendisch et al., 2017; Cohen et al., 2014). This study is part of





research activities to meet the scientific objectives of Collaborative Research Centers, CRC/Transregio 172 "ArctiC
Amplification: Climate Relevant Atmospheric and SurfaCe Processes, and Feedback Mechanisms (AC)³ (Wendisch
et al., 2017). A number of studies using global observations and climate models confirm this phenomenon, called
Arctic Amplification and provide evidence that it grows beyond the Arctic (Kim et al., 2017; Cohen et al., 2014).
Though, the attribution of the origins this phenomenon is controversial (Serreze et al., 2011; Pithan et al., 2014),
cloud cover is well-known to play a role in the Arctic surface-atmosphere radiation balance (Kellogg et al., 1975;
Curry et al., 1996). The accurate identification of Arctic clouds in the ground scenes of remote sensing
measurements made from space is therefore of intrinsic importance. However, cloud screening over the Arctic is a
challenging task. Since, all developed cloud detection methods encounter many obstacles originating from the
unique atmosphere and surface conditions in the Arctic (Curry et al., 1996). The Arctic clouds are mostly optically
thin and low with no remarkable contrast in commonly used visible or thermal or microwave measurements to the
underlying surface covered with highly reflecting snow and ice (Rossow and Garder 1993; Curry et al., 1996).
In addition to the importance of clouds to Arctic Amplification, errors in the identification of cloud in scene are
also one of the major sources of error in retrievals of a variety of data products for both satellite and ground-based
measurements at high latitude. For instance, the interference of cloud contamination in the Aerosol Optical
Thickness (AOT) retrieved by passive satellite remote sensing is a well-known issue (Shi et al., 2014; Várnai and
Marshak, 2015; Christensen et al., 2017; Arola et al., 2017). This limits the reliability and usefulness of the AOT
products in the assessment of the direct/ indirect impact of aerosols in Earth's energy balance in particular over the
Arctic. To avoid the uncertainty included in AOT products due to significant misclassification of heavy aerosol load
by thin clouds (which have similar reflectance properties) the development of an adequate cloud identification
algorithm is a prerequisite (Martins et al., 2002; Remer et al., 2012; Wind et al., 2016; Mei et al., 2017a, 2017b;
Christensen et al., 2017).
One recent approach to detect cloud-free snow and ice for aerosol retrieval over high latitudes used the spectral
shape of clear snow, ISTO (Istomina et al., 2010). The latter analyses the spectral behavior of each ground scene and
identifies clear snow or ice scenes from Advanced Along-Track Scanning Radiometer (AATSR) measurements.
Thresholds of the reflectance were empirically determined in seven spectral channels from the VIS to TIR. Defining
a reliable threshold which can guarantee a successful separation of cloud and cloud-free regions for the wide range
of atmospheric conditions and surface types is a challenging task. This is because of the similarity between spectral
reflectance of cloud and snow-ice (Lyapustin et al., 2008). In spite of progress made by this approach, adequate
discrimination of thin cloud above ice or snow is an inherent limitation of such threshold based techniques.
The European Space Agency (ESA) standard cloud product from AATSR is another example of an existing cloud
data product over the Arctic. This operational cloud mask is called the Synthesis of ATSR Data Into Sea-Surface
Temperature (SADIST) and is based on the latitudinal thresholds for various cloud types (Ghent et al., 2017). The
SADIST was initially developed for cloud screening over the ocean (Zavody et al., 2000). Birks et al. (2007)
modified this method to apply it over land. Later, Kolmonen et al. (2013) reported that the cloud flags included in
AATSR product are noticeably restricted and using this cloud product results in aerosol episodes not being observed.
Sobrino et al. (2016) reviewed different cloud clearing methods including the AATSR operational cloud mask in the





framework of Synergistic Use of The Sentinel Missions For Estimating And Monitoring Land Surface Temperature (SEN4LST) project and highlighted the potential uncertainty in different versions of this product, which result in these errors being propagated in subsequent data products. For example, the AATSR operational cloud mask falsely detects cloud in ~ 16 % of the observations. This is attributed to the flagging of land features (such as rivers) incorrectly as cloud (see Sobrino et al., 2013).

To avoid the uncertainty arising from the similarity of spectral characteristics of snow, ice and clouds, we decided to develop an algorithm based on a different strategy namely the use of time series measurements. The use of abrupt changes of TOA reflectance in time with the aim of cloud identification has been reported previously (Gómez-Chova et al., 2017; Lyapustin et al., 2008). An early example of this idea was proposed for low to middle latitudes by Rossow and Garder (1993) in the International Satellite Cloud Climatology Project (ISCCP). This method later evolved as a part of MultiAngle Implementation of Atmospheric Correction (MAIAC) algorithm (Lyapusitn et al., 2008), which is mainly designed for use with observations over land (low to middle latitudes), where the aim to simultaneously retrieve aerosol and surface properties. However, it has also been utilized by another study to identify snow grain size over Greenland (Lyapustin et al., 2009). Though, further optimization for the Arctic region is required and reported, a better performance in comparison to Moderate Resolution Imaging Spectroradiometer (MODIS) cloud mask is reported by Lyapustin et al. (2009).

The central assumption used in these algorithms for cloud identification, is that clear-sky reflectance is different to that of clouds, which exhibit high variation as a function of time (Lyapustin et al., 2008; Gómez-Chova et al., 2017). Knowledge of cloud-free scenes within a given time period, is achieved from knowledge of the variability of the measured TOA reflectance. Covariance analysis is used to estimate the spatial coherence. This has a long history in remote sensing studies using time series measurements (Leese et al., 1970; Lyapustin et al., 2008). The covariance computation assumes changes in the textural patterns of the observed scene, which originate from natural and man-made features such as topography, lakes or urban areas (Lyapustin et al., 2008). The use of the covariance analysis, which accounts for geometrical structures, minimizes issues originating from illumination variation and results in the same algorithm being applicable over both dark and bright surfaces (Lyapustin et al., 2008). For these reasons we decided to use Pearson Correlation Coefficient (PCC) as a function of covariance value for cloud detection over the Arctic. However, Lyapustin et al. (2008) reported that in spite of relative good performance, the covariance itself is not alone adequate for cloud identification in the case of homogenous surfaces or thin clouds. Therefore, we decided to use a combination of a PCC analysis and the reflectance of solar radiation at 3.7 μm. The latter utilizes the contrast between cloud and underlying surface making it possible to distinguish cloud-free snow and ice.

Another argument in favor of the use of time series analysis is the availability of multiple images by the AATSR and Sea and Land Surface Temperature Radiometer (SLSTR) sensor over the Arctic. For AATSR the revisit time of 3-4 days over mid-latitudes (Kolmonen et al., 2016) with more frequent at higher latitude which increase to 2 days over the Arctic (Soliman et al., 2012; Mei et al., 2013). For the two SLSTR it is 0.9 days at the equator (Coppo et al., 2010) with these values increasing at higher latitudes due to orbital convergence.

The AATSR/SLSTR Cloud Identification Algorithm (ASCIA) has been developed for use in the (AC)[3] project (Wendisch et al., 2017). The project aims to identify, investigate and evaluate parameters and feedback mechanisms





which contribute to Arctic amplification (Wendisch et al., 2017). Consequently, a long-term data record of AOT and cloud is required. It is planned to use the ASCIA to identify cloud free scenes for AOT retrieval. It is also planned to be applied to the observation by the SLSTR onboard Sentinel-3A and Sentinel-3B launched in 2016 and 2018 respectively which provide continuity of AATSR observations.

A full description of the new cloud identification and its application to AATSR data is presented in the following sections of this manuscript. First, a brief data description is presented in Sect. 2. The theory and methodology, used in our new ASCIA, are discussed in detail in Sect. 3. We evaluated the performance of the ASCIA by comparison of the cloud identification with that of the ESA standard cloud product for AATSR level2 nadir cloud flag while ASCIA is also applied to AATSR nadir observations, those obtained by applying ISTO, the MODIS cloud mask, the Surface synoptic observations (SYNOP) and the AErosol RObotic NETwork (AERONET). The results of the comparisons with five different source of cloud data are reported in Sect. 6. A discussion and set of conclusions, drawn from the study, are presented in Sect. 7.

## 2 Instruments and Data

### 2.1 AATSR data

The AATSR flown on board polar orbiting Envisat was primarily designed for measuring Sea Surface Temperature (SST) with accuracy higher than 0.3 k after ATSR-1 and ATSR-2 on European Remote Sensing-1, ERS-1 and ERS-2 (http://envisat.esa.int/handbooks/aatsr/CNTR.html). The AATSR delivered data from March 2002 until Envisat failed in 2012. The unique design of spectral coverage of AATSR enabled this sensor to measure reflected and emitted radiances in the VIS, (0.55 μm, 0.66 μm), NIR (0.87 μm, 1.6 μm) and three TIR channels (3.7 μm, 10.85 μm, 12.00 μm) with spatial resolution of $1{\times}1$ km$^2$ at nadir view and swath wide of 512 km. In Fig. 1 one example of the AATSR image over Svalbard is shown. It comprises three different wavelengths to highlight different information, which one can gain from the wide spectral coverage of this instrument. For example, in upper right panel in Fig. 1 the large drop of reflectance over snow/ice created a notable contrast between the cloud and the underlying surface at this wavelength in comparison to that found from the VIS channels used in the R(0.66 μm)G(0.87 μm)B(0.55 μm) image. A similar separation of snow/ice and cloud is observed in the reflectance at 3.7 μm shown in the lower left panel in Fig. 1. However, at the longer wavelength of 11 μm thin cloud patterns appear in the south-western scenes close to and above Svalbard, which have small signatures in the shorter wavelength. Combining the information from the different channels in an appropriate way enables the presence of cloud in the ground scenes to be accurately identified.

The conical imaging geometry of AATSR yields the dual viewing capability of this sensor. Each scene was imaged twice. The first measurement of the ground scenes is in the forward direction at a viewing angle of 55°. The second occurs 150 sec later at a near-nadir viewing angle. This capability is a design feature of AATSR to deliver an optimal and accurate atmospheric correction and thereby invert an accurate surface reflectance. The two views theoretically yield independent information about atmosphere and the surface to be retrieved. (http://envisat.esa.int/handbooks/aatsr/CNTR.html). The dual view approach intrinsically provides more information





than the single view for the study of surfaces with complex reflectance characteristics, such as snow and ice
(Istomina, 2012).
Examples of AOT algorithms applied to AATSR data are as follows: the AATSR Dual-View algorithm (ADV)
which was initially proposed by Veefkind et al. (1999) and AATSR single-view algorithm (ASV) by Veefkind et al.
(1998), the Swansea University (SU) algorithm (North et al., 1999) and Oxford RAL Aerosol and Cloud retrieval
(ORAC) algorithm (Thomas et al., 2009). These algorithms typically not optimized for the retrieval of AOT at high
latitudes. As the first task in delivering an algorithm, which delivers AOT at high latitude, the new ASCIA to
identify cloud and cloud free ground scenes has been applied to AATSR measurements.

## 2.2   SLSTR data

The SLSTR on-board Sentinel-3A was launched on the 16[th] February in 2016 as the successor of AATSR series to
provide the continuity of long term SST measurements. The Sentinel-3B satellite, which contains an identical
payload, was also launched by a Rockot/Breeze-KM launch vehicle from the Plesetsk Cosmodrome in northern
Russia, on the 25[th] April 2018. The design of the SLSTR instrument has some significant improvements with respect
to ATSR (Coppo et al., 2010). For example, the swath width of single view and dual view was increased from 500
km to 1420 km and 750 km respectively. This yields global revisit times of 1.9 days at equator for the dual view and
1 day for the single view. There are measurements of two additional channels in the SWIR, at the wavelengths of
1.37 μm and 2.25 μm, which are used to provide more accurate cloud, cirrus and aerosol information and used to
correct for atmospheric radiative transfer effects in the determination of surface reflectance (Coppo et al., 2010). The
Fig. 2 upper right panel shows the use of the new 1.37 μm measurements to detect thin cirrus clouds, which are only
weakly identified in reflectance at 3.7 μm shown in Fig. 2. As the radiance and TOA reflectance at this wavelength
are not measured by AATSR and because of currently unresolved calibration issues in SLSTR data, the current
design of ASCIA does not yet include 1.37 μm measurements. In addition, water vapor absorption above and within
clouds is considered as an obstacle in using this channel for cirrus detection (Meyer et al., 2010). Nevertheless, the
use of the measurements at this wavelength in thin cirrus detection should improve the performance of ASCIA in
future. SLSTR also has a higher spatial resolution of $0.5\times0.5$ km$^2$ in the VIS and SWIR measurements and two
channels dedicated to fire detection (Coppo et al., 2010). The use of the observations from SLSTR and AATSR
enables a long-term time series of clouds and aerosol parameters including AOT over the Arctic to be derived.

## 2.3   Data used in the cloud identification comparison studies

### 2.3.1   SYNOP

The SYNOP have been provided by World Meteorological Organization (WMO) with the purpose of mapping large
scale weather information around the world. However, the availability of the data is limited in the Arctic studies due
to the coverage of SYNOP stations in this region. For example, there are almost absent in central parts of the Arctic
Circle as is shown in Fig. 3. The SYNOP measurements made by an observer or automated fixed stations are
available in a standardized layout of numerical code which is called FM-12 by WMO (1995). The SYNOP reports
include a variety of meteorological parameters such as temperature, barometric pressure, visibility etc. as well as





cloud amount which are observed at synoptic hours simultaneously throughout the globe. We used SYNOP cloud fraction, which have a temporal resolution of 1-3 hours, to evaluate the performance of our new developed ASCIA over the Arctic region.

The use of SYNOP measurements to validate a cloud identification algorithm, or for that matter the cloud predicted by a climate model, the fact that the SYNOP cloud fraction is reported in okta scale, which ranges from 0 (completely clear sky) to 8 (completely obscured by clouds) has to be appropriately taken into account. Converting discrete okta values to continuous percentage ones has been done in different ways by climatologists. A common assumption is that 1 okta equals 12.5 % of cloud coverage (Boers et al., 2010; Kotarba, 2009). For use in this study it was necessary to make an estimate of the error or uncertainty in the okta in measurements. It is assumed that the man-made nature of cloudiness okta estimation have errors of ±1 okta and even larger values of ±2 okta in the non 0 or 8 okta situations (Boers et al., 2010; Werkmeister et al., 2015). Boers et al. (2010) suggested defining a larger range of 18.75 % for 1 okta instead of commonly used value of 12.5 %. We used this approach and defined percentage of cloud values for each okta, which are given in Table 1. More details about validation procedure are provided in Sect. 6.

### 2.3.2 AERONET

The AERONET is a network of approximately 700 ground-based sun photometers established by National Aeronautics and Space Administration (NASA) and PHOtométrie pour le Traitement Opérationnel de Normalisation Satellitaire (PHOTONS). This globally distributed network aims to provide long-term and continuous measurements of AOT, inversion products and perceptible water in diverse aerosol regimes (Holben et al., 1998). The high temporal resolution of 15 minutes, expected low accuracy of ~ 0.01 to 0.021 (Eck et al., 1999) as well as readily accessible public domain database provides a suitable dataset for aerosol research and characterization.

AERONET data are categorized and available in 3 levels: Level 1.0 (unscreened), Level 1.5 (cloud screened and quality controlled) and level 2.0 (quality assured). The data used in this work are selected from Level 1.5 to validate cloud identification results from newly developed ASCIA. More details of validation procedure are discussed in Sect. 6.

### 3 Theoretical background

### 3.1 Pearson Correlation Coefficient (PCC)

The PCC was proposed by Karl Pearson (1896) and is used in this study as an indicator of the correlation between sequential AATSR measurements. The PCC is also known as the Pearson Product-Moment Correlation Coefficient (PPMCC). It is a standard dimensionless statistical parameter commonly used to measure the strength and direction of the linear association between a pair of variables (Benesty et al., 2009). This parameter has extensively been used in many studies which pursue pattern analysis and recognition.

Our use of the PCC analysis is to separate the surface reflectance at a given viewing angle, which is stable over short time periods, from the cloud reflectance, which is highly variable over short time period. To describe the computational procedure developed here, let assume x, y as two random variables, then PCC can be written as a





function of covariance of x and y which is normalized by square root of their variances (Rodgers et al., 1988;
Benesty et al., 2009):

$$PCC = \frac{COV(x,y)}{\sigma_x \sigma_y}, \tag{1}$$

where COV(x,y) is the covariance of variables and σ is the root-mean-square variations of each random variables
(Rodgers et al., 1988; Benesty et al., 2009):

$$COV(x,y) = \frac{1}{N^2}\sum\sum(x_i - \bar{x})(y_i - \bar{y}) \text{ and } \sigma_x^2 = \frac{1}{N^2}\sum_{i=1}^{N}(x_i - \bar{x})^2, \tag{2}$$

$$PCC = \frac{\sum(x_i - \bar{x})(y_i - \bar{y})}{\left(\sum(x_i - \bar{x})^2 \; \sum(y_i - \bar{y})^2\right)^{1/2}}, \tag{3}$$

where $\bar{x}$ and $\bar{y}$ are the mean value of x and y variables respectively. The correlation coefficient parameter has values
between -1 and +1 (Rodgers et al., 1988). The PCC values were prepared in this study. The association between the
two variables is stronger if the absolute value is closer to 1, whereas if two variables are independent or in another
word "uncorrelated" PCC value will become 0 (Benesty et al., 2009). As a consequence of the above the PCC values
computed between several data pairs for ground scenes of the same area at different times provide an indication of
whether the scene is cloud covered or free of clouds.
For this aim, the use of all seven channels (0.55 μm, 0.66 μm, 0.87 μm, 1.6 μm, 3.7 μm, 11 and 12 μm) was
investigated. The visible channels (0.55 μm, 0.66 μm) on their own are not optimal to separate cloud free form
cloudy scenes, in particular for thin clouds. The SWIR and TIR such as 1.6 μm and beyond, where liquid water and
ice absorb provide useful information. There is a large reduction of reflectance between clear snow/ice as compared
to clouds between 0.87 μm and 1.6 μm (Kokhanovsky, 2006). Our routine takes advantage of this contrast through
the PCC calculation. One major contributors of error in aerosol retrieval is misclassifying heavy aerosol loads with
clouds. Using 1.6 μm reflectance which is less affected by aerosols than visible wavelengths addresses in part this
issue (Lyapustin et al., 2008).
A second question in PCC analysis (after wavelength selection) is definition of the optimal size of the block of
ground scene for PCC calculation. In early version of current algorithm, we set up $10\times10$ km$^2$ as the block size.
Since, aerosol retrieval would be carry out with the same spatial resolution. However, our investigations and
previous studies show that $10\times10$ km$^2$ is not sufficient to capture surface patterns. Thus, blocks of $25\times25$ km$^2$ area
as proposed in previous studies (Lyapustin et al., 2008) were used. The implementation of PCC analysis as used in
this study is discussed in more detail in Sect. 4.

**3.2   Reflectance of 3.7 μm thermal infrared channel**

The reflectance part of TIR Channels at 3.7 μm and 3.9 μm have been used in different studies to determine cloud
properties such as cloud effective radius and thermodynamic phase of the cloud or to discriminate cloud and





snow/ice covered surface (Meirink et al., 2016; Klüser et al., 2015; Musial et al., 2014; Khlopenkov, et al., 2007;
Pavolonis et al., 2005; Rosenfeld et al., 2004; Spangenberg et al., 2001; Allen et al., 1990). The reason for the wide
application of this channel in cloud identification methods is the difference in Single Scattering Albedo (SSA) at this
band compared to shorter VIS and INR wavelengths, which in turn results from the significant sensitivity of SSA to
thermodynamic phase and particle size of clouds (Platnick et al., 2008). For example, the scattering of liquid clouds,
having small droplets, is relatively larger than absorption and the ratio of NIR/VIS reflectance approaches 1 while in
the case of large liquid droplets or ice particles, the absorption increases and this ratio is closer to zero (Platnick et
al., 2008).
In addition, cloud-free snow reflects at a relatively weak level in comparison to clouds at 3.7 µm channel (Derrien
et al., 1993; Platnick et al., 2008). Therefore, the contrast due to different physical properties and radiance of
snow/ice and cloud at 3.7 µm makes the use of this channel advantageous for the identification of clouds. During
daytime, the measured Brightness Temperature (BT) at 3.7 µm is determined from the upwelling radiation which
comprises both reflected or scattered solar radiation and the thermal emission from the surface (Musial et al., 2014).
To use TOA reflectance at 3.7 µm, procedures are needed to account for and subtract the emission portion of
measured BT at 3.7 µm wavelength (Allen et al., 1990). To achieve this goal independent information about the
surface TIR is needed. This is estimated from observations at 11 µm where absorption by water vapor and other
trace gases is very small, most phenomena behave as blackbodies and the measured BT considered as the real
surface temperature (Istomina et al., 2010; Musial et al., 2014).
To do that, we use the method described in Meirink et al. (2016) and Musial et al. (2014), where the reflectance of
3.7 µm can be written as:

$$R_{3.7} = \frac{L_{3.7} - B_{3.7}(T_{11})}{\mu_0 F_{3.7,0} - B_{3.7}(T_{11})},$$    (4)

where $R_{3.7}$ is the reflectance i.e. the ratio of scattered radiance to incident solar radiance; L is measured radiance at
3.7 µm. The contribution from thermal emission at 3.7 µm is the Planck function radiance $B_{3.7}(T_{11})$ estimated from
the temperature value obtained from the measurements at 11 µm; $F_{3.7,0}$ is the solar constant at 3.7 µm which is
weighted by $\mu_0$ as the cosine of solar zenith angle.
Theoretical reflectance values at 3.7 µm band, computed by Allen et al. (1990) have been compared to satellite
measurements at the same channel from Advanced Very High ReSolution Radiometer (AVHRR). The results of this
work are summarized in Table 2. According to this study, the reflectance of liquid clouds primarily depends on
droplet size and solar zenith angle, whereas for ice clouds, ice particle shape and size distribution are of great
importance together with Cloud Optical Thickness (COT) and sun-satellite geometry. The observed reflectance is
reported in a range of 0.08 to 0.36 for liquid clouds and 0.02 to 0.27 for ice clouds (Allen et al., 1990). Arking and
Childs (1985) calculated 3.7 µm reflectance for ice clouds which varies between 0.01 to 0.30 for the COT of 0.1 to
100 and ice crystal effective radius of 2 µm to 32 µm, solar zenith angle of 60°. Spangenberg et al. (2001) reported a
typical value of 0.04 to 0.4 for clouds. In the case of snow covered surface 3.7 µm reflectance is dependent on many
factors including snow grain size, solar zenith angle, liquid water content, snow impurities and etc. Considering the





snow grain size of 50 μm to 200 μm, with a solar zenith angle of 40° to 80°, the modeled values for snow reflectance
varies between 0.005 and 0.025 at 3.7 μm (Allen et al., 1990). However, a range of 0.02 to 0.04 is observed from the
satellite measurements over the same wavelength for snow cover. This difference between model calculations and
measurements is explained by snow impurities (Allen et al., 1990). For land areas, the 3.7 μm reflectance is
impacted by soil type, vegetation type, coverage and moisture content. An average value of 0.15 is derived for clear
sky land scenes at 3.7 μm (Allen et al., 1990). In order to use the remarkable contrast between snow cover and
clouds at 3.7 μm channel, two main issues have to be taken into account: 1) the interference between snow and ice-
cloud values; 2) the interference between cloud and land reflectance. The latter is easily solved by using information
from visible channels with 3.7 μm reflectance. This is because land scenes are dark in comparison to cloud and
snow. The first issue, discriminating ice clouds from snow is a challenging task. To detect ice clouds, we combined
3.7 μm reflectance with PCC analysis. A full description of this new method is given in Sect. 4.

## 4   Methodology

The ASCIA implementation is initiated by preparing a time series of data. A time span of one month for the ground
scene was selected. Hagolle et al. (2015) indicated that in Sentinel-2 measurements with revisit time of 5 days, most
of the given scenes would be observed cloud-free at least once a month. Consequently, we also assume that every
scene of AATSR measurements, which have a higher revisit time of 3 days, will be cloud-free at least once a month.
Depending on the latitude and the time of year the number of downloaded data varies from 10 to 50 or more over
the same scene. AATSR provide more data over higher latitudes, which increase in spring and summer time due to
longer polar days and solar illumination. The AATSR L1b data are already provided as gridded and calibrated 1×1
km$^2$ scenes, which include geo-location information interpolated from the tie point scenes which are equally
distributed across a single AATSR image (http://envisat.esa.int/handbooks/aatsr/CNTR.html). Therefore, there is no
necessity to re-grid them for geo-referencing step which is considered as an advantage to preserve the original
reflectance value of each scene for following steps. However, the time series data are acquired by the satellite from
different viewing geometries. To compute PCC values over the same areas from different days, the ASCIA looks for
the closest similar scenes using geo-location information provided in the data. The closest distance is often found to
be within 0.006 degree and increases to 0.01 degree in the worst case and thus of negligible significance. After
finding the same blocks over different dates and building blocks, the ASCIA comprises two main parts: i) PCC
analysis at 1.6 μm; ii) Applying thresholds on reflectance of 3.7 μm channel.
In the first step, a PCC analysis for a block of ground scenes (25×25 km$^2$) is used to identify cloud and cloud-free
blocks which are assumed to have low and high PCC values respectively. The output of this step is a binary flag at
the block level. This serves as input for the second step to produce at ground scene level (1×1 km$^2$ or 0.5×0.5 km$^2$
depending on spatial resolution of instrument) cloud identification, by using the knowledge of the reflectance of
solar radiation at 3.7 μm channel. The combination of these two constraints is necessary because neither PCC
analysis nor reflectance part of 3.7 μm channel is adequate on its own for accurate cloud detection. A high PCC
value cannot guarantee the clearness of the whole block of scenes (Lyapustin et al., 2008) because some ground
scenes may still contain clouds, which are not enough in number to decrease significantly the PCC value. This case





occurs frequently over small or semitransparent clouds where the textural pattern of surface is still observable

through the clouds (Lyapustin et al., 2008). Small PCC values may be caused by a rapid surface change or high

aerosol load or the lack of recognizable spatial pattern, which is often the case over homogenous snow covered

surface (Lyapustin et al., 2008). A PCC value of 0.63 is suggested by Lyapustin et al. (2008) to separate cloud-free

blocks over middle latitudes. Considering less surface patterns in a large area of the Arctic compared to lower

latitudes, and our PCC analysis over both middle and high latitudes, we defined a lower threshold of PCC 0.4 over

the Arctic region and found the PCC of 0.6 is appropriate for middle latitudes based on large number of statistical

analysis.

After computing the first binary cloud flag at block level using last measurement and one previous image, the

ASCIA keeps the result in memory and repeats the procedure with second previous data and so on, until the last

measurement of the data series is involved in PCC analysis. The final binary blocks are imported through the second

step to identify cloudy scenes based on thresholds defined for blocks with low and high PCC value differently.

However, we would like to underline that, the snow/ice reflectance at 3.7 μm channel (~0.005-0.025) has

interference with those of ice clouds (0.01-0.3) at this wavelength. To avoid the uncertainty arising from this

problem we defined the PCC analysis as a decision point of ASCIA requiring further optimized analysis:

(i) For the high PCC $\geq$ 0.4, the whole block is considered to be cloud free and then the ASCIA starts looking for

remaining small cloud scenes within a block; scenes with $R_{3.7}$ larger than the maximum value observed over

snow at 3.7 μm: $R_{3.7} > 0.04$, (Allen et al., 1990).

(ii) For PCC < 0.4, the block is assumed to be cloudy; ASCIA removes all scenes within the block and only keeps

scenes which satisfy the $R_{3.7} < 0.015$ test. This threshold is equal or lower than the lowest observation of ice

cloud reflectance at 3.7 μm (Allen et al., 1990).

In our method, PCC analysis constrains the procedure and strict decision is only made within low PCC blocks.

The loss of some clear scenes in low PCC blocks is an unavoidable side effect of using strict criteria in particular

over land scenes, having low PCC and high 3.7 μm reflectance values. However, the ASCIA detects the presence of

thin cirrus cases with a relatively high confidence level. A schematic flowchart of the ASCIA is shown in Fig. 4,

with the use of the two main constraints being highlighted. In addition to picking out clear scenes, a simple land

classification procedure is undertaken in this step of the ASCIA. Snow/ice scenes are identified with low 3.7 μm

reflection whereas land scenes with high reflection are classified with the aid of the darkness test over visible

channels. The corresponding thresholds for land classification scheme are described in the Table 3.

It is important to note that if one scene, although characterized as land, may include soil, different types of

vegetation cover or even melting snow. The latter mix with soil and became dark enough to be filtered out from the

snow class. Sea-ice is distinguished from water on the basis of its higher brightness; one scene might be white

enough to be considered as ice. However, melting or broken ice would not be labeled as ice. Snow over sea-ice is

not distinguished from pure sea-ice and both of them are labeled as sea-ice. This also means that ice over land is also

marked as snow as well as pure snow.

A representative examples of the block level 25×25 km$^2$ and scene level 1×1 km$^2$ results of the ASCIA on

AATSR observations on the scenes within the region over northwest of Greenland in spring time enclosed in the



coordinates for four corners (75°N, 48°W), (75°N, 75°W), (81°N, 48°W), (81°N, 75°W) taken on the 18 May 2008
are shown in Fig. 5. This example selected to show the performance of ASCIA over a combination of fairly
homogenous snow cover, land, ocean, sea-ice and cloud. As we discussed earlier, the ambiguity of the PCC analysis
over homogenous surfaces on the right side of AATSR scene in Fig. 5, is entirely compensated by using additional
information from 3.7 μm channel. Another example over a surface with highly variable topography in March with
relatively higher solar zenith angle (>80°) is selected over Svalbard enclosed in the coordinates for four corners
(75°N, 4°E), (75°N, 32°E), (81°N, 4°E), (81°N, 32°E) taken on the 1 March 2008.

## 5  Results

In this study, we applied our recently developed ASCIA to identify cloud in the scenes using AATSR L1b (TOA
reflectance) and SLSTR L1b gridded data. The input file to the process chain is one scene of AATSR L1b product
the output comprises 5 classes of surface types including snow/ice, sea ice, water, cloud and land. The procedure of
surface classification is explained in Sect. 4. The location and time of selected case studies are used to show that the
identification of cloud by our new ASCIA is adequate. In this regard, the AATSR data are selected from several
years starting from 2006, during strong Arctic haze episode, which originated predominantly from agricultural fires
burning in Eastern Europe. The event has been reported in previously (Law et al., 2007). A second episode in 2008
is also considered for which validation data are available from SYNOP stations. One month of data from the
targeted seasons spring, summer and autumn vis. March, May, and July respectively have been acquired over
Greenland and Svalbard to assess the performance of the ASCIA in a wide range of solar zenith angles (60°-85°)
observed at high latitudes. In order to take into account various surface types in the Arctic, we selected case studies
including, highly variable topography and fairly homogenous snow cover, coast lines, land and ocean along snow
and ice covered surface. The design criteria for the ASCIA are optimized for an over various regions of the Arctic in
different solar illumination conditions with the exception of the dark winter period. The results obtained are
compared with i) the AATSR L2 nadir cloud flag and ii) those results obtained with ISTO (Istomina et al., 2010)
and iii) MODIS.
As we discussed in Sect. 1, misclassifying thin cirrus cloud with clear snow is reported as an unresolved problem
of ISTO approach. Two representative scenarios of this problem are illustrated in Fig. 6 and Fig. 7 over Greenland
and Svalbard respectively in which thin cloud is detected as clear snow by the ISTO method whereas ASCIA
confirmed the presence of cloud. In fact, over such a homogenous surface like Greenland, the second step of the
ASCIA plays the main rule. Since, the lack of structural patterns on surface lead to low PCC values in first step and
consequently overestimation of cloudy scenes. However, the reflection part of 3.7 μm could help to label and bring
back clear homogenous surface as cloud free snow in second step. The right panel in Fig. 6 and 7 shows the
difference between the result of ASCIA and ISTO. In this panel, the dark blue scenes show clouds which are not
detected by ISTO while the ASCIA could identify them sufficiently. On the other hand, reddish scenes show cloud
free scenes which ISTO failed to detect them but the ASCIA labeled them as cloud free. As we can see, in addition
to the edge of clouds which are difficult to detect specially over snow and ice, we have a remarkable number of





undetected cloud scenes in ISTO results which are identified successfully by the ASCIA. However, for the rest of these two scenes, both of two algorithms show a promising agreement.

The ESA cloud product from L2 data, shows a significant overestimation of clouds which leads to missing clear snow and ice scenes. The tendency of this product to flag clear scenes as cloud is also visible in Fig. 6, 7. The results in Fig. 8 show undetected clouds as another problem of AATSR level 2 cloud product, which happens frequently in winter time. To have a better understanding of this misclassification, we validated the AATSR L2 nadir cloud flag against SYNOP measurements and results are described in Sect. 6.

The lack of good performance in winter time over the Arctic with high solar zenith is observed in all of the results using ISTO method. Figure 8 is an example over Svalbard in March 2008. Over such a highly variable surface type like Svalbard, reflection part of 3.7 μm could approach the highest values such as 0.035, which is similar to that from cloud reflection. In this difficult case, PCC analysis is of great importance to keep cloud free snow scenes from the strict criteria of second step in particular in winter time with higher solar zenith angle. The ASCIA in high PCC block accounts a wide range of solar zenith angle (40-80 degree) and results in the reflectance of snow/ice being defined as between 0.02 and 0.04 at 3.7 μm channel. On the right panel, one can see the large number of red scenes which are falsely detected as cloud in the ISTO method.

Figure 10 shows one example of a haze event over Svalbard on 3$^{rd}$ of May, 2006. Both of ESA and ISTO cloud products had good performance for this case with the exception of the undetected thin cloud scenes which are falsely detected as clear snow by the ISTO. In fact, the appropriate design and application of PCC analysis over 1.6 μm enables cloud to be discriminated from heavy aerosol load. However, aerosol load over cloud could not be separated from cloudy scenes.

The only season, in which all three approaches detected clouds with similar success, was summer in July as shown in Fig. 9. Although ASCIA shows an overall better performance in particular for thin clouds, the required computational time for cloud detection and surface classification is higher than two other methods. In addition, we compared our results with those from the MODIS cloud identification algorithm used for masking clouded scenes. As an example, Fig. 11 shows the AATSR scene over Svalbard on 14$^{th}$ July 2008, where a large part of sea-ice is covered with thin clouds which have a small signature in visible channels. The middle panel shows the MODIS cloud mask for the same area. Although there is a small time difference of 15 minutes between MODIS and AATSR overpasses, we see that scenes identified with cloud by ASCIA correspond well with those of MODIS.

Figure 12 shows another example over northwest of Greenland on 18 May 2008. The thin and broken clouds are well detected over the snow cover by ASCIA, as well as the clouds over the southern part of the scene, which is covered with snow and ocean. As we can see from the comparison between ASCIA and MODIS cloud scene identification, cloudy scenes in the northern part of scene are not captured by MODIS product but the presence of clouds is seen in the RGB image in left panel. We observed other cases with similar differences especially for the case of thin and broken clouds. There are two potential sources of these differences, 1) time differences, which are 10 minutes in this case, or 2) a proper performance of the MODIS cloud mask over bright surfaces covered by snow and ice.





Due to the loss of Envisat and thus AATSR data in 2012, and the need for long time series of data, we tested
ASCIA on the AATSR successor SLSTR as well. Figure 13 shows some results over Svalbard on the 18[th] April
2017. Due to the smaller swath width of AATSR compared to SLSTR, the ASCIA is not applied to the full coverage
of SLSTR and the selected scene is cropped to have the similar coverage of 500×500 km$^2$. In spite of some
unresolved calibration issues in this sensor, the higher spatial resolution in SLSTR clearly helps to improve cloud
identification in first step, because the PCC analysis is more sensitive to smaller changes in 0.5×0.5 km$^2$ scenes
compared to 1×1 km$^2$. Moreover, the shorter revisit time of the Sentinel-3 satellite provides more acquired images
over the same scene. This results in a larger number of reference images, compared to those from Envisat. Overall
these effects result in the ASCIA application on SLSTR data being improved to the performance with AATSR.
However, the comparison of MODIS and ASCIA results indicates that ASCIA detected more cloudy scenes than the
MODIS algorithm.

## 6   Validation

In this section, we present a quantitative validation of our ASCIA results by making comparisons with simultaneous
ground-based SYNOP and AERONET measurements. The ESA standard cloud product is also compared with these
validation data sets. The difference in spatial and temporal resolution of the new cloud identification datasets and the
data sets used to validate this dataset has to be taken into account. The difference in the time of satellite and SYNOP
measurements is small being below ±20 minutes in most cases and generally does not exceed ± 45 minutes. To
compare surface measurement from SYNOP hemispheric view with the cloud identification at a spatial resolution
1×1 km$^2$ resolution satellite measurement, we calculated cloudiness as the percentage of cloudy scenes within a
window of 20×20 km$^2$ around each SYNOP station. This is a similar distance to that used in previous studies to
validate satellite based cloud identification SYNOP or similar surface measurements (Kotarba, 2017; Werkmeister
et al., 2015; Minnis et al., 2003). The cloud detection data product was then compared to the three months (March,
May and July) of SYNOP observations. These comprise 100 measurements over Svalbard and Greenland.
In Fig. 14 we present the relation between the calculated Cloud Fractional Cover (CFC) from ASCIA and SYNOP
measurements and density plot of occurrences of the CFC by ASCIA as a function of SYNOP following the idea of
Werkmeister et al. (2015). We find that these two data sets have a correlation coefficient of R=0.92. In 31 % of
scenarios, ASCIA estimates 1 okta more than SYNOP while in 14 % of match-ups SYNOP shows higher CFC of 1
okta. Figure 14 also reveals that most of ±1 okta differences occur when either SYNOP or ASCIA estimates 7 or 8
oktas which could be due to definition of 8 oktas (100 % CFC) and conversion of continuous percentage to okta
(Werkmeister et al., 2015). For instance, CFC of 99.9 % is considered as 7 oktas by using Table 1 while the CFC
difference is only 0.1 % with 8 oktas. The underestimation of CFC by SYNOP is also confirmed in the histogram of
difference between ASCIA-SYNOP in Fig. 15 which was indicated in previous studies as well (Kotarba, 2009;
Werkmeister et al., 2015). We also indicate the higher accuracy of ASCIA for cloud detection compared to ESA
cloud product. The results of the validation are summarized in Table 4. The cloud cover reported from SYNOP has
an overall agreement of 96 % (within ±2 okta) and 83 % (within ±1 okta) with cloud identification data from
ASCIA. As we discussed earlier error of ±1 to ±2 okta would be expected form SYNOP cloud cover values due to


man-made nature of its observation and viewing conditions (Boers et al., 2010; Werkmeister et al., 2015). The ESA
cloud product agrees 68 % (within ±2 okta) and 50 % (within ±1 okta) with SYNOP CFCs. The larger differences of
SYNOP and ESA cloud product are also indicated in Fig. 16 where the CFC values in percentage are shown for
ASCIA, ESA and SYNOP for validation scenarios. The blue error bars, indicate the range of okta values for each
SYNOP according to Table 1.
We also validated ASCIA cloud identification results with AERONET level 1.5 measurements. In 86.1 % of 36
studies scenes over Svalbard, both ASCIA and AERONET confirm the presence of clouds.

## 7   Conclusion

A new cloud detection algorithm, called ASCIA, for use at high altitudes above bright surfaces has been developed
to generate stand-alone products and for subsequent use in the retrieval of AOT over the Arctic. ASCIA uses data
from the European instruments using AATSR on ESA Envisat (2002 to 2012) and SLSTR on ESA Sentinel 3A or
3B. The ASCIA employs initially a time series analysis of PCC to identify cloud presence, the stability and cloud-
free conditions at the block scale of scenes (25×25 km$^2$). It then uses the 3.7 μm solar reflectance to discriminate
cloud presence at the spatial resolution of the scene level, which is 1x1 km$^2$ or 0.5×0.5 km$^2$ for AATSR and SLSTR
measurements respectively. The PCC parameter analysis is independent to a first approximation of threshold
settings, which lead to misclassification of cloud and snow due to the similarity of their spectral characteristics. The
brightness temperature measurements from 3.7 μm channel provide information to convert a block level (25×25
km$^2$) to a scene level (1×1 km$^2$ or 0.5×0.5 km$^2$) cloud identification. ASCIA thereby exploits the contrast in
reflectance between snow/ice and cloud at 3.7 μm wavelength.
The results of applying the new developed ASCIA are compared and validated against 5 existing products and
methods over the Arctic: 1) SYNOP measurements, 2) AERONET measurements, 3) one of existing methods based
on spectral shape of clear snow 4) AATSR L2 nadir cloud flag, 5) MODIS cloud product. The validation is resulted
in overall agreement of 96 % (within ±2 oktas) and 83 % (within ±1 okta) between SYNOP and ASCIA. The
comparison of the ASCIA and ISTO methods shows a better performance of ASCIA in extreme situations, such as
high solar zenith angle conditions.
The validation results indicate that the current ESA AATSR L2 nadir cloud flag often falsely identify clouds over
snow/ice with the exception of during summer. The comparison between ESA AATSR L2 cloud product and
SYNOP measurements resulted in 68 % (within ±2 oktas) and 50 % (within ±1 okta). The overall better
performance of ASCIA has also been shown by the SLSTR data. However, more investigation and optimization are
needed for the detection of cloud over land (soil, vegetation etc.) in the PCC blocks with lower values. Since, the
strict performance of the ASCIA in cloudy blocks results in scenes of clear land (without snow cover) being
identified as cloud due to high reflectance of land scenes at 3.7 μm channel. Additionally, sub-scene cloud detection
has not been studied with the current version of ASCIA. The use of reflectance in the 1.37 μm channel will be tested
in the future to improve thin cirrus detection in ASCIA.
*Acknowledgements: We gratefully acknowledge the support by the Collaborative Research Centres,*
*CRC/Transregio 172 "ArctiC Amplification: Climate Relevant Atmospheric and SurfaCe Processes, and Feedback*



*Mechanisms (AC)³. This work has been funded in part by the DFG CRC 172 and the State and University of*
*Bremen.*

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



**Table 1.** Calculation of cloudiness in percentage for corresponding okta values

| Percentage of Cloud | okta |
|:---:|:---:|
| 0 | 0 |
| $0 < \% < 18.75$ | 1 |
| $18.75 \leq \% < 31.25$ | 2 |
| $31.25 \leq \% < 43.75$ | 3 |
| $43.75 \leq \% < 56.25$ | 4 |
| $56.25 \leq \% < 68.75$ | 5 |
| $68.75 \leq \% < 81.25$ | 6 |
| $81.25 \leq \% < 100$ | 7 |
| 100 | 8 |





**Table 2.** Simulated and observed reflectance values at 3.7 µm (Allen et al., 1990)

| Surface/cloud Type | Simulation of 3.7 µm Reflectance | Observation of 3.7 µm Reflectance |
|---|---|---|
| Ice cloud | 0.01-0.3 | 0.02-0.27 |
| Liquid cloud | 0.1-0.45 | 0.08-0.36 |
| Clear land | ~0.15 | 0.03-0.1 |
| Snow cover | 0.005-0.025 | 0.02- 0.04 |




**Table 3.** Land classification criteria in cloud-free scene.

| Surface Type | Simulation of 3.7 μm Reflectance | Description |
| --- | --- | --- |
| Water | $R_{0.87} < 11\%$ & NDSI$\geq$0.4 | MODIS snow and ice mapping ATBD |
| Sea-ice | $R_{0.87} > 11\%$ & NDSI$\geq$0.4 | (Hall et al., 2001) |
| Land | $R_{3.7}>0.04$ & $R_{0.66} < 0.2$ \|\| NDSI<0.4 | Allen et al.,1999 |
| Snow | $R_{3.7}\leq0.04$ | Allen et al., 1999 |



**Table 4.** A summary of the comparison of ASCIA and ESA cloud product with SYNOP measurements used to validate these
products.

|  | Criteria | |
| --- | --- | --- |
| Cloud data | within ±2 oktas | within ±1 okta |
| ASCIA vs. SYNOP | 96 % correct<br>4 % incorrect | 83 % correct<br>17 % incorrect |
| ESA vs. SYNOP | 68 % correct<br>32 % incorrect | 50 % correct<br>50 % incorrect |





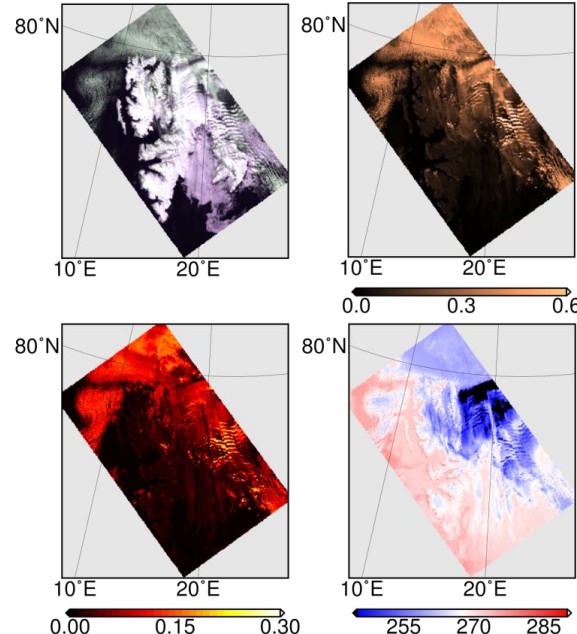

**Figure 1.** Upper left: the RGB image of AATSR over Svalbard, 10 May 2006, upper right: 1.6 μm reflectance, lower left: 3.7 μm reflectance, lower right: 11 μm brightness temperature.



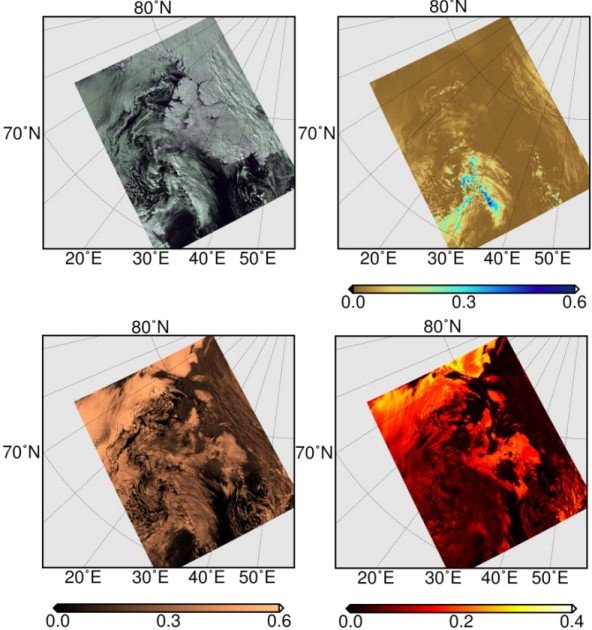

**Figure 2.** Upper left: the RGB image of SLSTR over Svalbard, 18 April 2017, upper right: 1.37 μm reflectance, lower left: 1.6 μm reflectance, lower right: 3.7 μm reflectance.



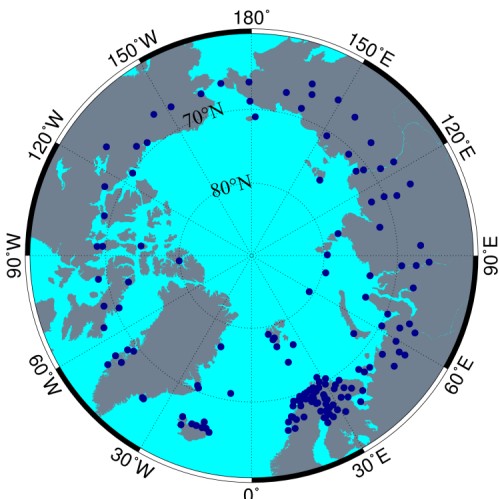

**Figure 3.** SYNOP network coverage over the Arctic, the dark blue points indicate the location of SYNOP stations.





**Figure 4.** The schematic flowchart of ASCIA.



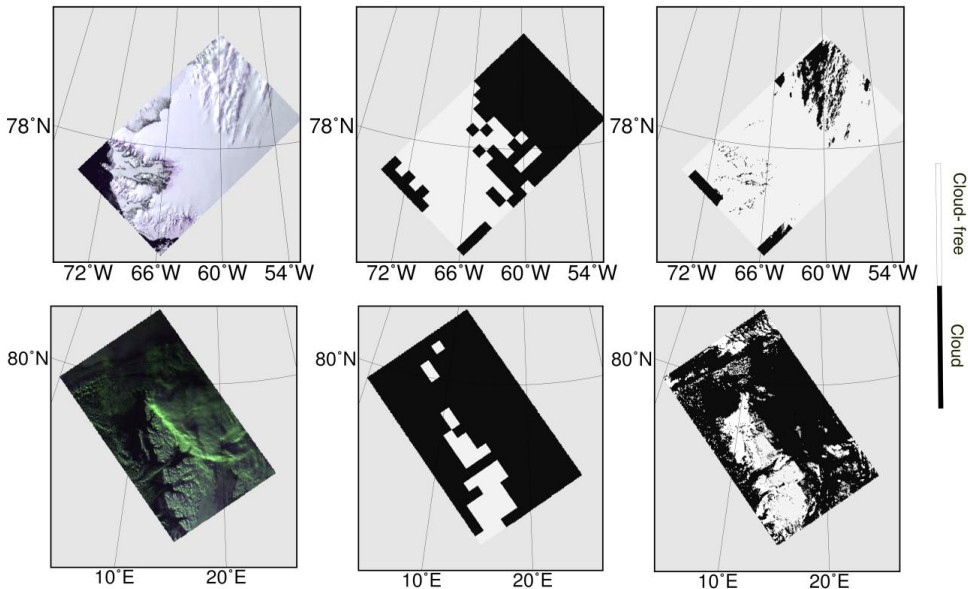

**Figure 5.** Examples of the results of ASCIA on AATSR observations on the scenes over Greenland (upper panels) and Svalbard (Lowe panels), taken on the 18 May 2008 and 1 March 2008 respectively, Left panels: RGB images, middle panels: Cloud detection at block level ($25\times25$ km$^2$), right panels: cloud detection at scene level.



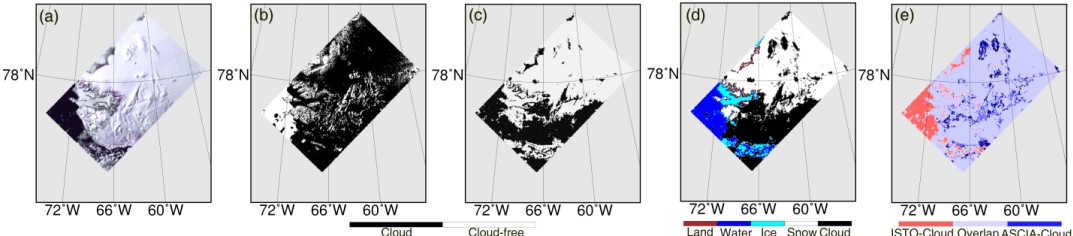

**Figure 6.** (a) The RGB image of AATSR over northern Greenland, 24 May 2008, (b) Nadir cloud flag from AATSR L2 product, (c) cloud detection based on spectral shape of clear snow, (d) cloud detection of ASCIA, (e) difference between ISTO and ASCIA

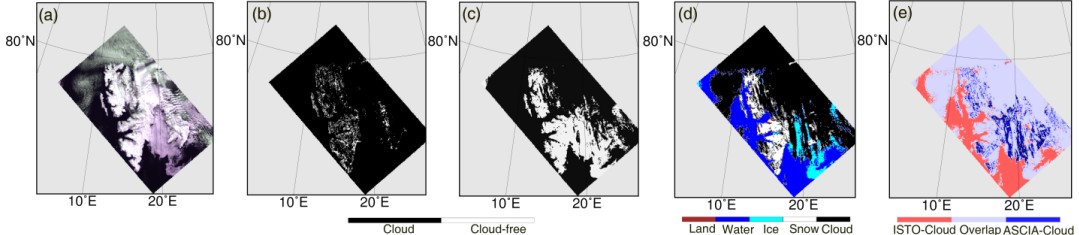

**Figure 7.** (a) The RGB image of AATSR over Svalbard, 10 May 2006, (b) Nadir cloud flag from AATSR L2 product, (c) cloud detection based on spectral shape of clear snow, (d) cloud detection of ASCIA, (e) difference between ISTO and ASCIA.

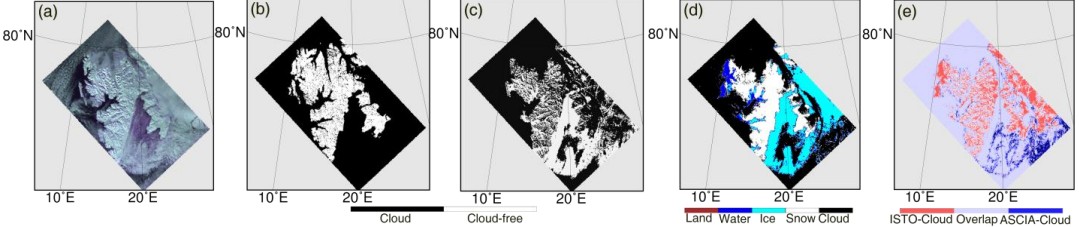

**Figure 8.** (a) The RGB image of AATSR over Svalbard, 18 March 2008, (b) Nadir cloud flag from AATSR L2 product, (c) cloud detection based on spectral shape of clear snow, (d) cloud detection of ASCIA, (e) difference between ISTO and ASCIA.



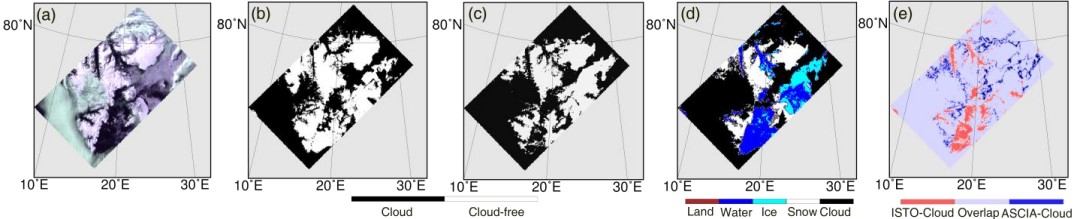

**Figure 9.** (a) The RGB image of AATSR over Svalbard, 6 July 2008, (b) Nadir cloud flag from AATSR L2 product, (c) cloud detection based on spectral shape of clear snow, (d) cloud detection of ASCIA, (e) difference between ISTO and ASCIA.

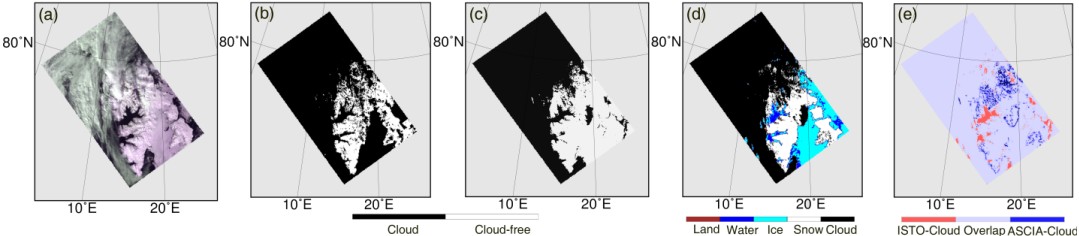

**Figure 10.** (a) The RGB image of AATSR over Svalbard, 3 May 2006, (b) Nadir cloud flag from AATSR L2 product, (c) cloud detection based on spectral shape of clear snow, (d) cloud detection of ASCIA, (e) difference between ISTO and ASCIA.





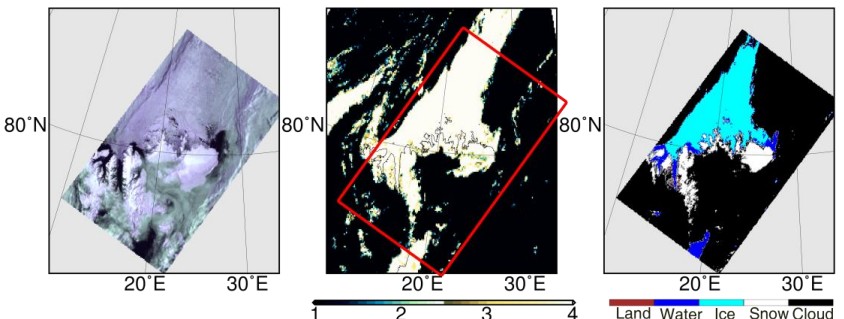

**Figure 11.** Left panel: RGB image of AATSR over Svalbard, 14 July 2008, 16h 40min 45s, middle panel MODIS cloud mask algorithm retrieved data: 1- cloudy, 2- probably cloudy, 3- probably clear, 4- clear, (red rectangle shows the coverage of AATSR) for 16h 25 min, right panel: the results for the cloud detection of ASCIA.

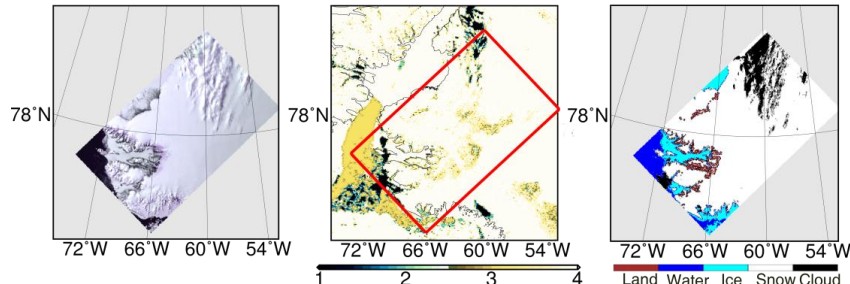

**Figure 12.** Left panel: RGB image of AATSR over Greenland, 18 May 2008, 23h 13min 38s, middle panel: MODIS cloud mask: 1- cloudy, 2- probably cloudy, 3- probably clear, 4- clear, (red rectangle shows the coverage of AATSR) for 23h 5min, right panel: Cloud detection of ASCIA.





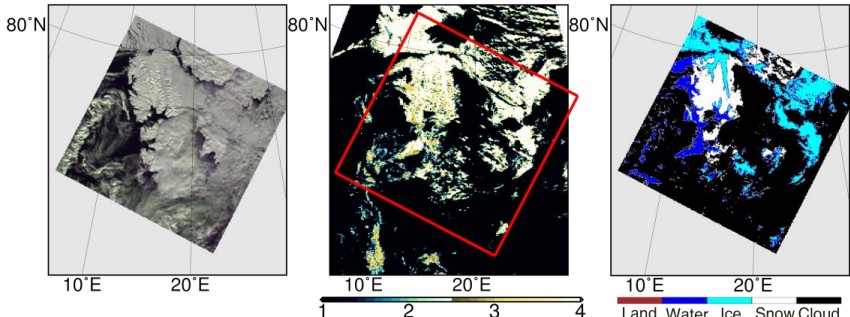

**Figure 13.** Left panel: The RGB image of SLSTR over Svalbard, 18 April 2017, 10hr 15min 6s, Middle panel: MODIS cloud

mask: 1- cloudy, 2- probably cloudy, 3- probably clear, 4- clear, (red rectangle shows the coverage of AATSR) for 11h 30m,

right panel: Cloud detection of ASCIA.




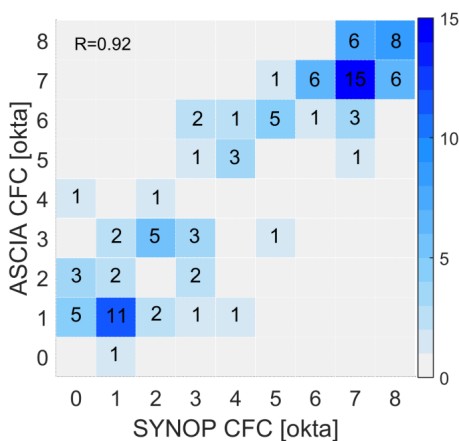

**Figure 14.** Density plot of occurrences of the CFC by ASCIA as a function of SYNOP.

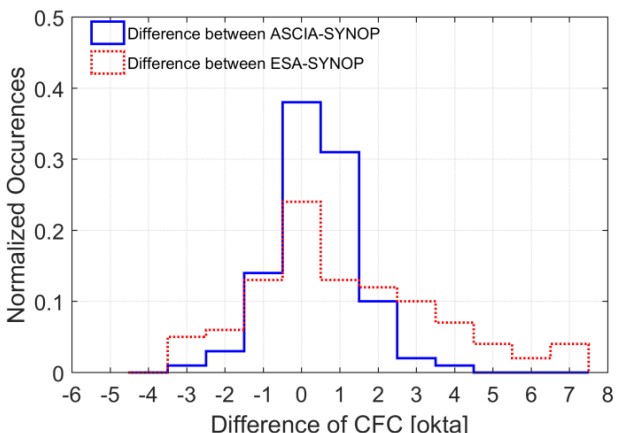

**Figure 15.** Histogram of CFC differences (blue: ASCIA minus SYNOP; red: ESA cloud product minus SYNOP).

674

675





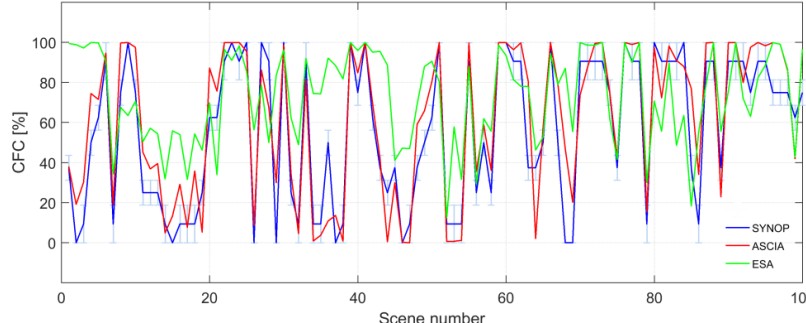

**Figure 16.** CFC in percent by ASCIA (red), SYNOP (blue) and ESA Cloud Product (green) for 100 scenarios of March, May and July 2008 over Svalbard and Greenland. Light blue error bars show the range of percentage values for each okta from SYNOP measurements.