# Peer review of "A cloud identification algorithm over the Arctic for use with AATSR/SLSTR measurements"

_Atmospheric Measurement Techniques, 2018_

## Referee Comment (RC1) · Anonymous Referee #1 · 14 Sep 2018

This paper identifies the need for good cloud detection algorithms over Polar Regions specifically for the purpose of aerosol retrievals, but this is also true of any atmospheric/surface retrieval in these areas. It is more difficult to identify cloud and classify the surface type in Polar Regions, and therefore the topic of this paper is of relevance to the scientific community. There are some things that I would like to see addressed before publication that I have included in the general comments section below. My technical comments are the more minor things that should be addressed such as typos.

General comments

1. Line 97: If you apply your algorithm to AATSR, clouds can change significantly in a period of 2 days quoted here as the return time. 2. Line 162: What about continuity in the data record? There is a ∼4 year gap between the failure of AATSR and the launch of SLSTR. 3. Line 250: The 11 micron BT although closest to surface temperature is not the 'real surface temperature' as it includes atmospheric effects. To give an example, LST or SST is never equal to BT in the 11 micron channel. If it were, we wouldn't need to retrieve surface temperature. 4. Line 312: Are you including all measurements within the block in the PCC calculation? This isn't clear in the text. If you are then your PCC is based on ∼25x25 pixels (x2 for two scenes?). I would think in this scenario that most of the information is coming from the spatial variability rather than the temporal variability? If you are using a block average, this seems like too few observations to make a valid PCC calculation. 5. Line 335: New ice is also dark and I think you would find it hard to distinguish from open water. It would be good to mention this here too. 6. Making comparisons with SYNOP using a 45-minute time window for validation could be problematic? Clouds can move across a scene within minutes? 7. *Please check the use of English throughout. The manuscript is readable but often the sentence structure isn't quite right and this makes it more difficult to understand. To give an example (line 222): o Manuscript: 'One major contributors of error in aerosol retrieval is misclassifying heavy aerosol loads with clouds.' o Correct English: 'One of the major contributors to error in aerosol retrievals is misclassification of heavy aerosol loads as cloud.' 8. The structure of the text within the various sections of the manuscript could be improved/tightened up in places, particularly in the introduction. 9. Figures 1-2, 5-13: It would be good to enlarge all of these figures so that they are more readable and cloud/surface features can be identified. 10. Figures 1-2, 5-13: It is far more intuitive for the reader if cloud is white and clear-sky is black. Colours should also match between algorithms. For example in one plot you have black=cloud, white=clear, and in another white=snow. This can become quite confusing for the reader to interpret. 11. Table 4: The use of the terms 'correct' and 'incorrect' here assumes that there is no uncertainty in the SYNOP data. Is this really true? How are the okta's determined

from this data? Does it involve human input? Is misclassification possible? I would be inclined to use the terms 'agreement' and 'disagreement' with the caveat that SYNOP isn't perfect made within the text. 12. Figures with RGB: Are these actual RGB's or false colour images? This should be made clear and the channels used described in the figure captions. 13. Figure 16: Light blue error bars are difficult to see. Could you change the colour? 14. There is some existing literature about cloud detection and surface classification over Polar Regions from within the SST CCI project: o Bulgin, C. E., Eastwood, S., Embury, O., Merchant, C. J. and Donlon, C. (2015) Sea surface temperature climate change initiative: alternative image classification algorithms for sea-ice affected oceans. Remote Sensing of Environment, 162. pp. 396-407. ISSN 0034-4257 doi: https://doi.org/10.1016/j.rse.2013.11.022 How does your algorithm compare to a Bayesian approach? This paper also shows the limitations of the SADIST mask that you mention in your paper. 15. Have you given any consideration to classification at nighttime? I realise that in the context of aerosol retrievals this isn't relevant, but it is for other applications so would be good to include a sentence on this.

Technical comments: 1. Line 32: Comma after 'though' doesn't make sense. 2. Line 36: Comma after 'since' doesn't make sense. Please check throughout, as there are not many instances in English where it is appropriate to use a comma after the first word in a sentence, except with 'however' on occasion. 3. Line 39: Snow/ice are also cold – this is the limiting factor in the thermal infrared. 4. Line 47: 'as' thin cloud? 5. Line 151/152: The revisit times stated here are not consistent with those given in the introduction (0.9 days). 6. Line 277: Only in Polar Regions – deserts for example can be bright. Sunglint over water is also bright. 7. Line 650: Should be 'lower'.

---

## Referee Comment (RC2) · Anonymous Referee #2 · 16 Dec 2018

This paper propose a new algorithm for cloud identification over the Artic area, which is a relevant topic for the scientific community. The results shown in the paper are promising, although the paper needs to be revised. Here are my major comments:

Please check you English. The content of the paper really suffer from lack of structure, wrong grammar and misplaced commas. Also, you should not use an article before ASCIA, except when it is used as adjective. Example: "ASCIA retrieves clouds over Artic" or "The ASCIA retrieval over Artic". This is true for all acronyms and abbreviations (e.g. SLSTR). Since (which certainly does not require a comma) introduces a subordinate sentence, which cannot be separated from the main clause by a full stop.

It is not clear to me whether or not your algorithm is applicable during winter. At line 357 you write that your targeted seasons are spring, summer and autumn and then you choose March, May and July. This is already confusing by itself. Later, at line 362 you write that ASCIA is not optimized for winter time. Could you please clarify this?

Fig. 5 The left panel over Greenland shows 2 rectangles in left part of the image. Could you please discuss where they arise from? Your algorithm shows promising results, but it is always worthwhile to discuss its limitations.

L398 You say the computing time is higher. How higher? Please give an estimate.

L428 I agree with the other reviewer, 45 minute time difference seem to me quite large for validation purposes. Maybe you should introduce a filtering?

Sections Results and Validation could be compressed in one section, as even when presenting the results you do some qualitative validation against other cloud products.

L454 Could you please show part of the evaluation against AERONET? AS the latter is a well-known reference for every reader, the validation against it deserves more than 2 lines of text. Also, which version are you using? And why L1.5 instead of L2.0?

Technical comments L151 The SLSTR revisit time is 1.9 day at the equator with one satellite and 0.9 day with two satellites, not single/dual view.

Table 3. The title of the second column should be something like "Test"

Figures 1-2, fig. from 5 to 13 and fig. 16 should indeed be larger.

L339-347 please simplify these lines. Throughout your manuscript sentences are often too long, but here they really affect the readability. Simplify the lines here and maybe add more information on the caption of the figures (for example the exact coordinates of the corners).

L16 reflection at 3.7 um
L18 e,g. e.g.

L32 Though the attribution of the origins of this phenomenon

L76 the aim is

L104 it is also planned to apply it to the observations acquired by SLSTR

L123 In the upper right

L131 Each scene is

- L142 These algorithms are typically not optimized
- L168 For example, they are almost absent in the central parts

L187 AERONET is ...

- L206 Then the PCC can be written as
- L207 function of the covariance

L241 no new line

L254 Here you should compact everything in one sentence

L285 AATSR provides

L310 found that a PCC of 06

L319 ASCIA starts looking for remaining small cloud scenes within a block, i.e. scenes  $\ldots$  (R3.7 >0.04)

- L333 it is important to note that one scene, ...
- L334 The latter mix with soil and becomes
- L339 a representative example

L373 cloud free scene which ISTO failed to detect but are correctly labeled by ASCIA.
**L393 Both the ESA and ISTO**

L447 would be expected from SYNOP

AMTD

---

## Author Comment (AC1) · 27 Jan 2019

We would like to thank you for the review and your constructive comments which help to improve the manuscript. Please find our detailed responses in the supplement.

Best regards, Soheila

Please also note the supplement to this comment:
https://www.atmos-meas-tech-discuss.net/amt-2018-231/amt-2018-231-AC1-supplement.pdf

---

## Author Comment (AC2) · 27 Jan 2019

**Reply to Anonymous Referee #2**

We would like to thank you for the review and your constructive comments which help to improve the manuscript. We have revised the manuscript accordingly and our point-by-point responses (in blue) to the specific comments (in red) are given. The modification made in the manuscript is presented in green. This document also includes a marked-up version of manuscript.

Best Regards, Soheila Jafariserajehlou

Comments to the Author:

Reviewer #2: Please check you English. The content of the paper really suffer from lack of structure, wrong grammar and misplaced commas. Also, you should not use an article before ASCIA, except when it is used as adjective. Example: "ASCIA retrieves clouds over Artic" or "The ASCIA retrieval over Artic". This is true for all acronyms and abbreviations (e.g. SLSTR). Since (which certainly does not require a comma) introduces a subordinate sentence, which cannot be separated from the main clause by a full stop.

✓ We have corrected all these points throughout the manuscript.

Reviewer #2: It is not clear to me whether or not your algorithm is applicable during winter. At line 357 you write that your targeted seasons are spring, summer and autumn and then you choose March, May and July. This is already confusing by itself. Later, at line 362 you write that ASCIA is not optimized for winter time. Could you please clarify this?

- ✓ We are sorry for this inaccurate statement. The algorithm is not optimized for night time retrievals (the information from the visible channels may not be valid during night) and we do not attempt to apply it during polar night. In fact, we only had access to 3 months (March, May and July) of SYNOP data which we used for validation. We understood the confusion arising from this statement and it would be better not to mention seasons here. We modified the text as below:
- ✓ Modification: Line 380 388: Three months of data from March, May and July have been acquired over Greenland and Svalbard to assess the performance of ASCIA in a wide range of solar zenith angles (60°-85°), surface and atmospheric conditions observed at high latitudes. In order to take various surface types in the Arctic into account, we selected case studies including, highly variable topography and fairly homogeneous snow cover, coast lines, land and ocean along snow and ice covered surface. The designed criteria for ASCIA are optimized for various regions over the Arctic observed under different solar illumination conditions. Polar night and transition seasons at low light conditions are excluded from our retrievals.

Reviewer #2: Fig. 5 The left panel over Greenland shows 2 rectangles in left part of the image. Could you please discuss where they arise from? Your algorithm shows promising results, but it is always worthwhile to discuss its limitations.

 $\checkmark$  We see these rectangles on the middle and right panels over Greenland:

**Middle panel:** In the middle panel, these 2 rectangles are indicators of the blocks with low Pearson Correlation Coefficient (PCC smaller than 0.4). As we can see from the corresponding RGB image, the locations of these two rectangles are over/near open water. We already discussed this shortcoming of the PCC analysis at lines 322-325 (in the original version of this manuscript): "Small PCC values may be caused by rapid surface change, high aerosol load or lack of recognizable spatial pattern, which is often the case over homogeneous snow covered surface". Therefore the lack of geo-physical patterns and structure could be a potential reason in this case. We also mentioned at lines 318-319 in the original version of manuscript "The combination of these two constraints is necessary because neither PCC analysis nor the reflectance part of 3.7  $\mu$  m is adequate on its own for accurate cloud detection".

**Right panel:** However, using the reflectance of 3.7  $\mu$  m compensates the limitation of the PCC criterion as we explained at line 393-396. As we can see in the final results of this image (in Fig. 12 in the results section) these two rectangles over open water disappeared. Therefore the right panel in Fig. 5 (in original version of the manuscript) should not have these rectangles as well.

We thank the referee for this information. We realized that we have used erroneously an outdated version of this picture (which was not created using the right running version of the algorithm).

We have updated the right panel in Fig. 5 (with an improved color scheme (comment from the reviewer #1)). Please see below.

✓ Modification: Line 705 Fig. 5.

Reviewer #2: L398 You say the computing time is higher. How higher? Please give an estimate.

✓ The run-time of one scene is 30 minutes on a state-of-the-art computer system.

Reviewer #2: L428 I agree with the other reviewer, 45 minute time difference seem to me quite large for validation purposes. Maybe you should introduce a filtering?

- ✓ In preparation of the article, we performed a comprehensive review of previous works to define the optimal maximum time difference. However, statements in literature strongly vary: 10 min (Werkmeister et al., 2015), 15 min (Musial et al., 2014), 1 h (Dybbroe et al., 2005) and 4 h (Meerkötter et al., 2004), obviously all for different kinds of meteorological conditions. The investigation and results in the previous publications indicate that temporal difference in validation of satellite retrievals against SYNOP may vary based on meteorological conditions. Therefore we also tried to check meteorological conditions in our study.
- ✓ First, we checked to see which fraction of our results could be affected by longer temporal difference? We found that in 30% of our validation scenarios time difference exceeds half an hour. We tried to answer your question in these 30%. Do clouds change in the interval of 45 minutes? Or clouds could be almost stable due to the unique Arctic environment compared to lower latitudes? To answer this question we have performed 2 investigations:

1) We checked SYNOP data before and after validation time to see whether cloud changes could be observed during this time? Available to us were SYNOP data, i.e. cloud fraction every 3 hours over Svalbard and every 1 hour over Greenland. We found that only in 12% of our validation scenarios cloud fraction changed ( $\pm 1$  or  $\pm 2$  oktas) within 2 hours (for Greenland) and – 6 hours (for Svalbard).

2) Another perspective on this topic is to ask how fast cloud can travel during our validation time window? We have checked the wind speed over Svalbard to answer the question. How strong the wind needs to be to move clouds out of or into in our validation area which is defined as a circle with the radius of 20 km around each SYNOP station? If we assume that the cloud is in the middle of this circle, we need at least a wind speed of 7.5 m/s to move clouds out of or into this circle in 45 minutes. Based on information from the Norwegian Meteorological Institute which provides an average of hourly wind speed, we see that the average wind speed over or close to the selected SYNOP stations at the closest stations during satellite overpass time is usually very low. For example below 3m/s and only in one scenario, it exceeds 7.5m/s slightly.

- Second, choosing a smaller temporal difference like for instance 0.5h would limit the number of observations and introduce a sampling error. For example, by filtering the validation dataset with respect to 0.5h temporal difference, 30% of the validation data will be lost. Therefore, to have a trade-off between good statistics/sampling and representativeness, we decided to keep the temporal interval of 45 minutes.
- ✓ However, temporal difference between satellite and SYNOP measurements is one of several sources of uncertainty (different viewing perspective, different spatial footprint etc.) which affect validation results.

We have updated the text to address this comment and explain the uncertainty which originates from time difference.

- ✓ Modification: Line 453-465: To define the optimal maximum temporal difference between SYNOP and satellite data, other comparable validation activities used different temporal intervals like 10 min (Werkmeister et al., 2015), 15 min (Musial et al., 2014), 1 h (Dybbroe et al., 2005) and 4 h (Meerkötter et al., 2004). The investigation and results in the previous publications indicate that temporal difference in validation of satellite retrievals against SYNOP may vary based on meteorological conditions. Allowing only a small temporal difference between measurement datasets (here: SYNOP and ASCIA) ensures as close as possible temporal resemblance but can introduce a significant sampling error due to limited number of validation cases (Bojanowski et al., 2014). According to Bojanowski et al. (2014) a temporal difference of 90 min to compare with SYNOP measurements at temporal difference will introduce an error which should be considered along other sources of uncertainty (different viewing perspective, different spatial footprint and etc.). In this study, the maximum allowed temporal difference between the ASCIA retrievals and SYNOP measurements is small being below ±20 minutes in most cases and generally does not exceed ± 45 minutes.
- ✓ Line 544: Bojanowski, J., Stöckli, R., Tetzlaff, A., and Kunz, H.: The Impact of Time Difference between Satellite Overpass and Ground Observation on Cloud Cover Performance Statistics, Remote Sensing, 6, 12 866–12 884, https://doi.org/10.3390/rs61212866, 2014.
- Line 561: Dybbroe, A., Karlsson, K.-G., Thoss, A., NWCSAF AVHRR cloud detection and analysis using dynamic thresholds and radiative transfer modeling. Part II: Tuning and validation. J. Appl. Meteorol., 44, 55–71, 2005.
- ✓ Line 626: Meerkötter, R., König, C., Bissolli, P., Gesell, G., Mannstein, H., A 14-year European cloud climatology from NOAA/AVHRR data in comparison to surface observations. Geophys. Res. Lett., 31, doi:10.1029/2004GL020098, 2004.

Reviewer #2: Sections Results and Validation could be compressed in one section, as even when presenting the results you do some qualitative validation against other cloud products.

✓ Done.

Reviewer #2: L454 Could you please show part of the evaluation against AERONET? AS the latter is a well-known reference for every reader, the validation against it deserves more than 2 lines of text. Also, which version are you using? And why L1.5 instead of L2.0?

✓ We agree and modified the text as below. Unfortunately it was not possible to perform further statistical analysis (like we did for SYNOP) in validation against AERONET because the comparison is pixel-based and does not include a circular area around the station where we could estimate cloud fraction.

The reason for selecting level 1.5 instead of level 2.0 is that, level 1.5 data are cloud screened but level 2.0 data are quality assured. This means if we use level 1.5 data and check whether we have aerosol

measurements or not (having aerosol measurements means cloud-free condition) we could have information of cloudiness. But, in level 2.0 data, missing aerosol data could also be due to low quality of measurements.

Modification: Line: 492-495: We also validated ASCIA cloud identification results with AERONET level 1.5 measurements. Because level 1.5 data are cloud screened. The procedure of this validation includes 2 steps: (1) covering AERONET observed AOT to a cloud flag (AOT is provided in AERONET only in cloud-free conditions); (2) Validation of ASCIA with AERONET cloud flag. In 86.1 % of 36 studied scenes over Svalbard, both ASCIA and AERONET confirm the presence of clouds.

Reviewer #2: Technical comments L151 The SLSTR revisit time is 1.9 day at the equator with one satellite and 0.9 day with two satellites, not single/dual view.

- ✓ Done.
- ✓ Modification: lines 161-162: This yields global revisit times of 1.9 days at the equator with two satellites and 0.9 day with one satellite.

| Reviewer #2: Table 3. The title of the second | d column should be something like "Test" |
|-----------------------------------------------|------------------------------------------|
|-----------------------------------------------|------------------------------------------|

- ✓ Done.
- ✓ Modification: line 698.

| Surface Type | Test                                                              | Description                     |
|--------------|-------------------------------------------------------------------|---------------------------------|
| Water        | $R_{0.87} < 11\% \& \text{NDSI} \ge 0.4$                          | MODIS snow and ice mapping ATBD |
| Sea-ice      | $R_{0.87} > 11\%$ & NDSI $\ge 0.4$                                | (Hall et al., 2001)             |
| Land         | $R_{3.7}\!\!>\!\!0.04$ & $R_{0.66}<0.2\parallel\text{NDSI}{<}0.4$ | Allen et al.,1999               |
| Snow         | $R_{3.7} \le 0.04$                                                | Allen et al., 1999              |

Reviewer #2: Figures 1-2, fig. from 5 to 13 and fig. 16 should indeed be larger.

✓ Done.

Reviewer #2: L339-347 please simplify these lines. Throughout your manuscript sentences are often too long, but here they really affect the readability. Simplify the lines here and maybe add more information on the caption of the figures (for example the exact coordinates of the corners)

- ✓ Done.
- ✓ Modification: line 357-362: A representative example of the block level (25×25 km2) and scene level (1×1 km2) results of ASCIA applied to AATSR observations is shown in Fig. 5. This example was selected to show the performance of the algorithm in presence of different surface conditions: 1) one scene is over a combination of fairly homogeneous snow cover, land, ocean, sea-ice and cloud scene at north-west of

Greenland taken on the 18 May 2008; 2) another example is over a surface with highly variable topography over Svalbard with relatively higher solar zenith angle (> $80^{\circ}$ ) on the 1 March 2008. As we discussed earlier, the ambiguity of the PCC analysis over homogeneous surfaces on the right and left side of the AATSR scene in middle panel of Fig. 5, is compensated in the right panel by using additional information from 3.7 µm channel.

✓ Line 707-711: Figure 5. Examples of the results of ASCIA on AATSR observations on the scenes over Greenland (upper panels) between (75°N, 48°W), (75°N, 75°W), (81°N, 48°W), (81°N, 75°W), taken on the 18 May 2008 and Svalbard (lower panels) within (75°N, 4°E), (75°N, 32°E), (81°N, 4°E), (81°N, 4°E), (81°N, 32°E) (lower panels), on the 1 March 2008, Left panels: RGB images, middle panels: Cloud detection at block level (25×25 km2), right panels: cloud detection at scene level.

**Reviewer #2: L16 reflection at 3.7 um**

- ✓ Done.
- ✓ Modification: line 16-17: Subsequently, the reflection at 3.7  $\mu$  m is used for accurate cloud identification at scene level at either 1×1 km2 or 0.5×0.5 km2.

**Reviewer #2: L18 e,g. e.g.**

- ✓ Done.
- ✓ Modification: line 17-18: The ASCIA data product has been validated by comparison with independent observations e.g. surface synoptic observations (SYNOP)

**Reviewer #2: L32 Though the attribution of the origins of this phenomenon**

- ✓ Done.
- ✓ Modification: line 32: Though the attribution of the origins of this phenomenon is controversially discussed.

**Reviewer #2: L76 the aim is**

- ✓ Done.
- ✓ Modification: line 79-80: ... where the aim is to simultaneously retrieve aerosol and surface properties.

**Reviewer #2: L104 it is also planned to apply it to the observations acquired by SLSTR**

- ✓ Done.
- ✓ Modification: line 111-112: It is also planned to apply it to the observations acquired by SLSTR onboard Sentinel-3A and Sentinel-3B launched in 2016 and 2018 respectively which provide continuity of AATSR observations.

**Reviewer #2: L123 In the upper right**

✓ Done.

✓ Modification: line132-133: For example, in the upper right panel in Fig. 1 the large drop of reflectance over snow/ice created a notable contrast...

**Reviewer #2: L131 Each scene is**

- ✓ Done.
- ✓ Modification: line 140-141: Each scene is imaged twice.

**Reviewer #2: L142 These algorithms are typically not optimized**

 $\checkmark$  Done. We deleted this line.

**Reviewer #2: L168 For example, they are almost absent in the central parts**

- ✓ Done.
- ✓ Modification: line 183-184: For example, they are almost absent in the central parts of the Arctic Circle as is shown in Fig. 3.

**Reviewer #2: L187 AERONET is ...**

- ✓ Done.
- ✓ Modification: line 203-205: AERONET is a network of approximately 700 ground-based sun photometers established by National Aeronautics and Space Administration (NASA) and PHOtométrie pour le Traitement Opérationnel de Normalisation Satellitaire (PHOTONS).

**Reviewer #2: L206 Then the PCC can be written as**

- ✓ Done.
- ✓ Modification: line 222-223: To describe the computational procedure developed, we assume x, y to be two random variables, then the PCC can be written as...

**Reviewer #2: L207 function of the covariance**

- ✓ Done.
- ✓ Modification: line 223: ...a function of the covariance of x and y which is normalized by square root of their variances.

**Reviewer #2: L241 no new line**

- ✓ Modification: line 259.
- ✓ Done.

**Reviewer #2: L254 Here you should compact everything in one sentence**

- ✓ Done.
- ✓ Modification: line 273: where  $R_{3.7}$  is the reflectance i.e. the ratio of scattered radiance to incident solar radiance; L measured radiance at 3.7 µm,  $B_{3.7}(T_{11})$  the Planck function radiance (the contribution from

thermal emission at 3.7  $\mu$ m) T11 measurements at 11  $\mu$ m; F3.7,0 the solar constant at 3.7  $\mu$ m and  $\mu_0$  the cosine of solar zenith angle.

**Reviewer #2: L285 AATSR provides**

- ✓ Done.
- ✓ Modification: line 304: AATSR provides more data over higher latitudes, which increase in spring and summer time due to longer polar days and solar illumination.

**Reviewer #2: L310 found that a PCC of 06**

- ✓ Done.
- ✓ Modification: line 327-329: ...we defined a lower threshold for PCC of 0.4 over the Arctic region and found that a PCC of 0.6 is appropriate for middle latitudes based on a number of statistical analyses.

Reviewer #2: L319 ASCIA starts looking for remaining small cloud scenes within a block, i.e. scenes . . . (R3.7 > 0.04)

- ✓ Done.
- ✓ Modification: line 338: ASCIA starts looking for remaining small cloud scenes within a block, i.e. scenes with R3.7 ...

**Reviewer #2: L333 it is important to note that one scene, . . .**

- ✓ Done.
- ✓ Modification: line 351: It is important to note that one scene,...

**Reviewer #2: L334 The latter mix with soil and becomes**

- ✓ Done.
- ✓ Modification line 352-353: The latter mix with soil and becomes dark enough to be filtered out from the snow class. Sea-ice is distinguished from water on the basis of its greater brightness...

**Reviewer #2: L339 a representative example**

- ✓ Done.
- ✓ Modification: line 357: A representative example of the block level 25×25 km2 and scene level 1×1 km2 results of ASCIA...

**Reviewer #2: L373 cloud free scene which ISTO failed to detect but are correctly labeled by ASCIA.**

- ✓ Done.
- ✓ Modification: line 398-399: On the other hand, reddish scenes show cloud free case which ISTO fails to detect but are correctly labeled by ASCIA as cloud free.

**Reviewer #2: L393 Both the ESA and ISTO**

✓ Done.

✓ Modification line 416: Both the ESA and ISTO cloud products showed good results for this case with the exception of undetected thin cloud scenes which are falsely labeled as clear snow by ISTO.

**Reviewer #2: L447 would be expected from SYNOP**

- ✓ Done.
- ✓ Modification: line 486-487: As discussed earlier an error of ±1 to ±2 okta would be expected as the accepted accuracy range from SYNOP cloud cover values due to man-made nature of its observation and viewing conditions

**A cloud identification algorithm over the Arctic for use with AATSR/SLSTR measurements**

Soheila Jafariserajehlou1, Linlu Mei1, Marco Vountas1, Vladimir Rozanov1, John P. Burrows FRS1, Rainer Hollmann2

1Institute of Environmental Physics, University of Bremen, Otto-Hahn-Allee 1, Bremen, 28359, Germany 2DWD – Deutscher Wetterdienst, Frankfurter Straße 135, 63067 Offenbach, Germany

Correspondence to: Soheila Jafariserajehlou (jafari@iup.physik.uni-bremen.de)

Abstract. The accurate identification of the presence of cloud in the ground scenes observed by remote sensing satellites is an end in itself. Our lack of knowledge of cloud at high latitudes increases the error and uncertainty in the evaluation and assessment of the changing impact of aerosol and cloud in a warming climate. A prerequisite for the accurate retrieval of Aerosol Optical Thickness, AOT, is the knowledge of the presence of cloud in a ground scene.

In this study observations of the up welling radiance in the visible (VIS), near infrared (NIR), shortwave infrared 6 (SWIR), and the thermal infrared (TIR), coupled with solar extraterrestrial irradiance are used to determine the 7 reflectance. We have developed a new cloud identification algorithm for application to the reflectance observations 8 of Advanced Along-Track Scanning Radiometer (AATSR) on European Space Agency (ESA)-Envisat and Sea and 9 Land Surface Temperature Radiometer (SLSTR) onboard the ESA Copernicus Sentinel-3A and -3B. The resultant 10 AATSR/SLSTR Cloud Identification Algorithm (ASCIA) developed addresses the requirements for the study AOT 11 at high latitudes and utilizes time-series measurements. It is assumed that cloud free surfaces have unchanged or 12 little changed patterns for a given sampling period, whereas cloudy or partly cloudy scenes show much higher 13 variability in space and time. In this method, the Pearson Correlation Coefficient (PCC) parameter is used to 14 measure the 'stability' of the atmosphere-surface system observed by satellites. The cloud free surface is classified 15 by analyzing the PCC values at the block scale  $25 \times 25$  km2. Subsequently, the reflection at  $\frac{1}{2}$  3.7 µm is used for 16 accurate cloud identification at the scene level at either  $1 \times 1 \text{ km}^2$  or  $0.5 \times 0.5 \text{ km}^2$ . The ASCIA data product has been 17 validated by comparison with independent observations e.g. ssurface synoptic observations (SYNOP), AErosol 18 RObotic NETwork (AERONET) and the following satellite-products from i) ESA standard cloud product from 19 AATSR L2 nadir cloud flag, ii) one method based on clear-snow spectral shape developed at IUP Bremen (Istomina 20 et al., 2010), which we call, ISTO, iii) Moderate Resolution Imaging Spectroradiometer (MODIS). In comparison to 21 ground based SYNOP measurements, we achieved a promising agreement better than 95 % and 83 % within ±2 and 22  $\pm 1$  okta respectively. In general, ASCIA shows an improved performance in comparison to other algorithms applied 23 to AATSR measurements for cloud identification at high latitudes. 24

**25 **1 Introduction**

[revised manuscript text omitted]

For this aim, the use of all seven channels (0.55  $\mu$ m, 0.66  $\mu$ m, 0.87  $\mu$ m, 1.6  $\mu$ m, 3.7  $\mu$ m, 11 and 12  $\mu$ m) was investigated. The visible channels (0.55  $\mu$ m, 0.66  $\mu$ m) on their own are not optimal to separate cloud free form cloudy scenes, in particular for thin clouds. The SWIR and TIR such as 1.6  $\mu$ m and beyond, where liquid water and ice absorb provide useful information. There is a large reduction of reflectance between clear snow/ice as compared

to clouds between 0.87  $\mu$ m and 1.6  $\mu$ m (Kokhanovsky, 2006). Our routine takes advantage of this contrast through

the PCC calculation. One major contributors of error in aerosol retrieval is misclassifying heavy aerosol loads with

239 clouds One of the major contributors to error in aerosol retrievals is misclassification of heavy aerosol loads as

240 cloud. Using 1.6 μm reflectance which is less affected by aerosols than visible wavelengths addresses in part this

issue (Lyapustin et al., 2008).

A second question in PCC analysis (after wavelength selection) is definition of the optimal size of the block of ground scene for PCC calculation. In early version of current algorithm, we set up  $10 \times 10$  km2 as the block size. Since, aerosol retrieval would be carry out with the same spatial resolution. However, our investigations and previous studies show that  $10 \times 10$  km2 is not sufficient to capture surface patterns. Thus, blocks of  $25 \times 25$  km2 area as proposed in previous studies (Lyapustin et al., 2008) were used. The implementation of PCC analysis as used in this study is discussed in more detail in Sect. 4.

**248 **3.2** Reflectance of 3.7 μm thermal infrared channel**

The reflectance part of TIR Channels at 3.7 µm and 3.9 µm have been used in different studies to determine cloud 249 properties such as cloud effective radius and thermodynamic phase of the cloud or to discriminate cloud and 250 snow/ice covered surface (Meirink et al., 2016; Klüser et al., 2015; Musial et al., 2014; Khlopenkov, et al., 2007; 251 Pavolonis et al., 2005; Rosenfeld et al., 2004; Spangenberg et al., 2001; Allen et al., 1990). The reason for the wide 252 application of this channel in cloud identification methods is the difference in Single Scattering Albedo (SSA) at this 253 band compared to shorter VIS and INR wavelengths, which in turn results from the significant sensitivity of SSA to 254 thermodynamic phase and particle size of clouds (Platnick et al., 2008). For example, the scattering of liquid clouds, 255 having small droplets, is relatively larger than absorption and the ratio of NIR/VIS reflectance approaches 1. But 256 while in the case of large liquid droplets or ice particles, the absorption increases and this ratio is closer to zero 257 (Platnick et al., 2008). 258

-In addition, cloud-free snow reflects at a relatively weak level in comparison to clouds at 3.7 µm channel (Derrien 259 et al., 1993; Platnick et al., 2008). Therefore, the contrast due to different physical properties and radiance of 260 snow/ice and cloud at 3.7 µm makes the use of this channel advantageous for the identification of clouds. During 261 daytime, the measured Brightness Temperature (BT) at 3.7 µm is determined from the upwelling radiation which 262 comprises both reflected or scattered solar radiation and the thermal emission from the surface (Musial et al., 2014). 263 To use TOA reflectance at 3.7 µm, procedures are needed to account for and subtract the emission portion of 264 measured BT at 3.7 µm wavelength (Allen et al., 1990). To achieve this goal independent information about the 265 surface TIR is needed. This is estimated from observations at 11 µm where absorption by water vapor and other 266 trace gases is very smalli, most objects phenomena in regions outside of the tropics can be treated behave as 267 blackbodies and the measured BT considered being in good agreement with as the real surface temperature 268 (Istomina et al., 2010; Musial et al., 2014). 269

To do that, we use the method described in Meirink et al. (2016) and Musial et al. (2014), where the reflectance of 3.7 μm can be written as:

$$R_{3.7} = \frac{L_{3.7} - B_{3.7}(T_{11})}{\mu_0 F_{3.7,0} - B_{3.7}(T_{11})},\tag{4}$$

272 273

274

275

where  $R_{3.7}$  is the reflectance i.e. the ratio of scattered radiance to incident solar radiance; L-is measured radiance at 3.7  $\mu$ m2. T\_B3.7(T11)\_the Planck function radiance (the contribution from thermal emission at 3.7  $\mu$ m) is the Planck function radiance B3.7(T11)=
[revised manuscript text omitted]
 ( $25 \times 25 \text{ km}^2$ ) and scene level ( $1 \times 1 \text{ km}^2$ ) results of the ASCIA 357 applied toon AATSR observations is shown in Fig. 5. This example was selected to show the performance of 358 ASCIA in presence of different surface conditions: 1) one scene is over a combination of fairly homogenous snow 359 cover, land, ocean, sea-ice and cloud scene at north-west of Greenland taken on the 18 May 2008; 2) another 360 example is over a surface with highly variable topography over Svalbard with relatively higher solar zenith angle 361 (>80°) on the 1 March 2008. on the scenes within the region over northwest of Greenland in spring time enclosed in 362 the coordinates for four corners (75°N, 48°W), (75°N, 75°W), (81°N, 48°W), (81°N, 75°W) taken on the 18 May 363 2008 are shown in Fig. 5. This example selected to show the performance of ASCIA over a combination of fairly 364 homogenous snow cover, land, ocean, sea ice and cloud. As we discussed earlier, the ambiguity of the PCC analysis 365 over homogeneous surfaces on the right and left side of the AATSR scene in middle panel of Fig. 5, is entirely 366 compensated in the right panel by using additional information from 3.7 µm channel. Another example over a 367 surface with highly variable topography in March with relatively higher solar zenith angle (>80°) is selected over 368 Svalbard enclosed in the coordinates for four corners (75°N, 4°E), (75°N, 32°E), (81°N, 4°E), (81°N, 32°E) taken on 369
- 370 the 1 March 2008.
- 371 **5 Results and validation**

**372 **5.1 The comparison to space-borne products**

In this study, we applied our recently developed ASCIA to identify cloud in the scenes using AATSR L1b (TOA reflectance) and SLSTR L1b gridded data. The input file to the process chain is one scene of AATSR L1b product the output comprises 5 classes of surface types including snow/ice, sea ice, water, cloud and land. The procedure of

surface classification is explained in Sect. 4. The location and time of selected case studies are used to show that the 376 identification of cloud by our new ASCIA is adequate. In this regard, the AATSR data are selected from several 377 years starting from 2006, during strong Arctic haze episode, which originated predominantly from agricultural fires 378 burning in Eastern Europe. The event has been reported in previously (Law et al., 2007). A second episode in 2008 379 is also considered for which validation data are available from SYNOP stations. ThreeOne months of data from the 380 targeted seasons spring, summer and autumn vis. March, May, and July respectively have been acquired over 381 Greenland and Svalbard to assess the performance of the ASCIA in a wide range of solar zenith angles  $(60^\circ - 85^\circ)_A$ 382 surface and atmospheric conditions observed at high latitudes. In order to take into account-various surface types in 383 the Arctic take into account, we selected case studies including, highly variable topography and fairly homogeneous 384 snow cover, coast lines, land and ocean along snow and ice covered surface. The designed criteria for the ASCIA 385 are optimized for an over various regions overof the Arctic observed underin different solar illumination conditions. 386 Polar night and transition seasons at low light conditions are excluded from our retrievals. 
[revised manuscript text omitted]
. To define the optimal maximum temporal 453 difference between SYNOP and satellite data, other comparable validation activities used different temporal 454 intervals like 10 min (Werkmeister et al., 2015), 15 min (Musial et al., 2014), 1 h (Dybbroe et al., 2005) and 4 h 455 (Meerkötter et al., 2004). The investigation and results in the previous publications indicate that temporal difference 456 in validation of satellite retrievals against SYNOP may vary based on meteorological conditions. Allowing only a 457 small temporal difference between measurement datasets (here: SYNOP and ASCIA) ensures as close as possible 458 temporal resemblance but can introduce a significant sampling error due to limited number of validation cases 459 (Bojanowski et al., 2014). According to Bojanowski et al. (2014) a temporal difference of 90 min to compare with 460 SYNOP measurements at temporal resolution of 3 h minimizes the sampling error (Bojanowski et al., 2014). 461 However, potential longer temporal difference will introduce an error which should be considered along other 462 sources of uncertainty (different viewing perspective, different spatial footprint and etc.). In this study, the 463 maximum allowed temporal difference between the ASCIA retrievals and SYNOP measurements is small being 464 below  $\pm 20$  minutes in most cases and generally does not exceed  $\pm 45$  minutes. The difference in the time of satellite 465 and SYNOP measurements is small being below ±20 minutes in most cases and generally does not exceed ± 45 466 minutes. To compare surface measurement from SYNOP hemispheric view with the cloud identification at a spatial 467 resolution 1×1 km2 resolution satellite measurement, we calculated cloudiness as the percentage of cloudy scenes 468 within a window of  $20 \times 20$  km2 around each SYNOP station. This is a similar distance to that used in previous 469 studies to validate satellite based cloud identification SYNOP or similar surface measurements (Kotarba, 2017; 470 Werkmeister et al., 2015; Minnis et al., 2003). The cloud detection data product was then compared to the three 471 months (March, May and July) of SYNOP observations. These comprise 100 measurements over Svalbard and 472 Greenland. 473

[revised manuscript text omitted]

in the future to improve thin cirrus detection in ASCIA. Regarding the aim of this work, night time cloud 523

- identification is not considered in the current version of ASCIA. For further applications, new criteria will be added 524
- to identify clouds during night time. 525

Acknowledgements: We gratefully acknowledge the support by the Collaborative Research Centres, 526 CRC/Transregio 172 "ArctiC Amplification: Climate Relevant Atmospheric and SurfaCe Processes, and Feedback 527 Mechanisms (AC)3. This work has been funded in part by the DFG CRC 172 and the State and University of 528 Bremen. 529

References 530

- Allen, R. C., Durkee, P. A., and Wash, C. H.: Snow/cloud discrimination with multispectral satellite measurements, 531 J. Appl. Meteor., 29, 994-1004, 1990. 532
- Arking, A. and Childs, J. D.: Retrieval of cloud cover parameters from multispectral satellites images, J. Climate 533 Appl. Meteor., 24, 322-333, 1985.
- Arola, A., Eck, T. F., Kokkola, H., Pitkanen, M. R. A., Romakkaniemi, S.: Assessment of cloud-related fine-mode 535 AOD enhancement based on AERONET SDA product, Atmos. Chem. Phys., 17, 5991-6001, 2017. 536
- Benesty, J., Chen, J., Huang, Y., Cohen, I.: Noise Reduction in Speech Processing, Springer Topics in Signal 537 Processing 2, Springer-Verlag Berlin Heidelberg, doi: 10.1007/978-3-642-00296-0\_5, 2009. 538
- Birks, A. R.: Improvements to the AATSR IPF relating to Land Surface Temperature Retrieval and Cloud Clearing 539 over Land, AATSR Technical Note, Rutherford Appleton Laboratory, Chilton, UK, 2007. 540
- Boers R., de Haij, M.J., Wauben, W. M. F., Baltink, H. K., van Ulft, L. H., Savenije, M., Long, C. N.: Optimized 541
- fractional cloudiness determination from five ground-based remote sensing techniques, J. Geophys. Res., 115, 542 D24116, 2010. 543
- Bojanowski, J., Stöckli, R., Tetzlaff, A., and Kunz, H.: The Impact of Time Difference between Satellite Overpass 544 and Ground Observation on Cloud Cover Performance Statistics, Remote Sensing, 6, 12 866-12 884, 545 https://doi.org/10.3390/rs61212866, 2014. 546
- Bulgin, C. E., Eastwood, S., Embury, O., Merchant, C. J. and Donlon, C., Sea surface temperature climate change 547 initiative: alternative image classification algorithms for sea-ice affected oceans. Remote Sens. Environ, 162, pp. 548 396-407, 2015. 549
- Christensen, M. W., Neubauer, D., Poulsen, C. A., Thomas, G. E., McGarragh, G. R., Povey, A. C., Proud, S. R., 550 Grainger, R. G.: Unveiling aerosol-cloud interactions - Part 1: Cloud contamination in satellite products 551 enhances the aerosol indirect forcing estimate, Atmos. Chem. Phys., 17, 13151-13164, 2017. 552
- Cohen, J., Screen, J. A., Furtado, J., C., et al.: Recent Arctic amplification and extreme mid-latitude weather, Nat. 553 Geosci., 7(9), 627

---

## Author Comment (AC3) · 27 Jan 2019

**Reply to Anonymous Referee #2**

We would like to thank you for the review and your constructive comments which help to improve the manuscript. We have revised the manuscript accordingly and our point-by-point responses (in blue) to the specific comments (in red) are given. The modification made in the manuscript is presented in green. This document also includes a marked-up version of manuscript.

Best Regards,
Soheila Jafariserajehlou
* * *
Comments to the Author:
Reviewer #2: Please check you English. The content of the paper really suffer from lack of structure, wrong grammar and misplaced commas. Also, you should not use an article before ASCIA, except when it is used as adjective. Example: "ASCIA retrieves clouds over Artic" or "The ASCIA retrieval over Artic". This is true for all acronyms and abbreviations (e.g. SLSTR). Since (which certainly does not require a comma) introduces a subordinate sentence, which cannot be separated from the main clause by a full stop.

✓ We have corrected all these points throughout the manuscript.

Reviewer #2: It is not clear to me whether or not your algorithm is applicable during winter. At line 357 you write that your targeted seasons are spring, summer and autumn and then you choose March, May and July. This is already confusing by itself. Later, at line 362 you write that ASCIA is not optimized for winter time. Could you please clarify this?

✓ We are sorry for this inaccurate statement. The algorithm is not optimized for night time retrievals (the information from the visible channels may not be valid during night) and we do not attempt to apply it during polar night. In fact, we only had access to 3 months (March, May and July) of SYNOP data which we used for validation. We understood the confusion arising from this statement and it would be better not to mention seasons here. We modified the text as below:

✓ Modification: Line 380 - 388: Three months of data from March, May and July have been acquired over Greenland and Svalbard to assess the performance of ASCIA in a wide range of solar zenith angles (60°-85°), surface and atmospheric conditions observed at high latitudes. In order to take various surface types in the Arctic into account, we selected case studies including, highly variable topography and fairly homogeneous snow cover, coast lines, land and ocean along snow and ice covered surface. The designed criteria for ASCIA are optimized for various regions over the Arctic observed under different solar illumination conditions. Polar night and transition seasons at low light conditions are excluded from our retrievals.

Reviewer #2: Fig. 5 The left panel over Greenland shows 2 rectangles in left part of the image. Could you please discuss where they arise from? Your algorithm shows promising results, but it is always worthwhile to discuss its limitations.

✓ We see these rectangles on the middle and right panels over Greenland:

**Middle panel:** In the middle panel, these 2 rectangles are indicators of the blocks with low Pearson Correlation Coefficient (PCC smaller than 0.4). As we can see from the corresponding RGB image, the locations of these two rectangles are over/near open water. We already discussed this shortcoming of the PCC analysis at lines 322-325 (in the original version of this manuscript): "Small PCC values may be caused by rapid surface change, high aerosol load or lack of recognizable spatial pattern, which is often the case over homogeneous snow covered surface". Therefore the lack of geo-physical patterns and structure could be a potential reason in this case. We also mentioned at lines 318-319 in the original version of manuscript "The combination of these two constraints is necessary because neither PCC analysis nor the reflectance part of 3.7 $\mu$ m is adequate on its own for accurate cloud detection".

**Right panel:** However, using the reflectance of 3.7 $\mu$ m compensates the limitation of the PCC criterion as we explained at line 393-396. As we can see in the final results of this image (in Fig. 12 in the results section) these two rectangles over open water disappeared. Therefore the right panel in Fig. 5 (in original version of the manuscript) should not have these rectangles as well.

We thank the referee for this information. We realized that we have used erroneously an outdated version of this picture (which was not created using the right running version of the algorithm).

We have updated the right panel in Fig. 5 (with an improved color scheme (comment from the reviewer #1)). Please see below.

✓ Modification: Line 705 Fig. 5.

[Figure]

Reviewer #2: L398 You say the computing time is higher. How higher? Please give an estimate.

✓ The run-time of one scene is 30 minutes on a state-of-the-art computer system.

Reviewer #2: L428 I agree with the other reviewer, 45 minute time difference seem to me quite large for validation purposes. Maybe you should introduce a filtering?

✓ In preparation of the article, we performed a comprehensive review of previous works to define the optimal maximum time difference. However, statements in literature strongly vary: 10 min (Werkmeister et al., 2015), 15 min (Musial et al., 2014), 1 h (Dybbroe et al., 2005) and 4 h (Meerkötter et al., 2004), obviously all for different kinds of meteorological conditions. The investigation and results in the previous publications indicate that temporal difference in validation of satellite retrievals against SYNOP may vary based on meteorological conditions. Therefore we also tried to check meteorological conditions in our study.

✓ First, we checked to see which fraction of our results could be affected by longer temporal difference? We found that in 30% of our validation scenarios time difference exceeds half an hour. We tried to answer your question in these 30%. Do clouds change in the interval of 45 minutes? Or clouds could be almost stable due to the unique Arctic environment compared to lower latitudes? To answer this question we have performed 2 investigations:

1) We checked SYNOP data before and after validation time to see whether cloud changes could be observed during this time? Available to us were SYNOP data, i.e. cloud fraction every 3 hours over Svalbard and every 1 hour over Greenland. We found that only in 12% of our validation scenarios cloud fraction changed (±1 or ±2 oktas) within 2 hours (for Greenland) and – 6 hours (for Svalbard).

2) Another perspective on this topic is to ask how fast cloud can travel during our validation time window? We have checked the wind speed over Svalbard to answer the question. How strong the wind needs to be to move clouds out of or into in our validation area which is defined as a circle with the radius of 20 km around each SYNOP station? If we assume that the cloud is in the middle of this circle, we need at least a wind speed of 7.5 m/s to move clouds out of or into this circle in 45 minutes. Based on information from the Norwegian Meteorological Institute which provides an average of hourly wind speed, we see that the average wind speed over or close to the selected SYNOP stations at the closest stations during satellite overpass time is usually very low. For example below 3m/s and only in one scenario, it exceeds 7.5m/s slightly.

✓ Second, choosing a smaller temporal difference like for instance 0.5h would limit the number of observations and introduce a sampling error. For example, by filtering the validation dataset with respect to 0.5h temporal difference, 30% of the validation data will be lost. Therefore, to have a trade-off between good statistics/sampling and representativeness, we decided to keep the temporal interval of 45 minutes.

✓ However, temporal difference between satellite and SYNOP measurements is one of several sources of uncertainty (different viewing perspective, different spatial footprint etc.) which affect validation results.

We have updated the text to address this comment and explain the uncertainty which originates from time difference.

✓ Modification: Line 453-465: To define the optimal maximum temporal difference between SYNOP and satellite data, other comparable validation activities used different temporal intervals like 10 min (Werkmeister et al., 2015), 15 min (Musial et al., 2014), 1 h (Dybbroe et al., 2005) and 4 h (Meerkötter et al., 2004). The investigation and results in the previous publications indicate that temporal difference in validation of satellite retrievals against SYNOP may vary based on meteorological conditions. Allowing only a small temporal difference between measurement datasets (here: SYNOP and ASCIA) ensures as close as possible temporal resemblance but can introduce a significant sampling error due to limited number of validation cases (Bojanowski et al., 2014). According to Bojanowski et al. (2014) a temporal difference of 90 min to compare with SYNOP measurements at temporal resolution of 3 h minimizes the sampling error (Bojanowski et al., 2014). However, potential longer temporal difference will introduce an error which should be considered along other sources of uncertainty (different viewing perspective, different spatial footprint and etc.). In this study, the maximum allowed temporal difference between the ASCIA retrievals and SYNOP measurements is small being below ±20 minutes in most cases and generally does not exceed ± 45 minutes.

✓ Line 544: Bojanowski, J., Stöckli, R., Tetzlaff, A., and Kunz, H.: The Impact of Time Difference between Satellite Overpass and Ground Observation on Cloud Cover Performance Statistics, Remote Sensing, 6, 12 866–12 884, https://doi.org/10.3390/rs61212866, 2014.

✓ Line 561: Dybbroe, A., Karlsson, K.-G., Thoss, A., NWCSAF AVHRR cloud detection and analysis using dynamic thresholds and radiative transfer modeling. Part II: Tuning and validation. J. Appl. Meteorol., 44, 55–71, 2005.

✓ Line 626: Meerkötter, R., König, C., Bissolli, P., Gesell, G., Mannstein, H., A 14-year European cloud climatology from NOAA/AVHRR data in comparison to surface observations. Geophys. Res. Lett., 31, doi:10.1029/2004GL020098, 2004.

Reviewer #2: Sections Results and Validation could be compressed in one section, as even when presenting the results you do some qualitative validation against other cloud products.

✓ Done.

Reviewer #2: L454 Could you please show part of the evaluation against AERONET? AS the latter is a well-known reference for every reader, the validation against it deserves more than 2 lines of text. Also, which version are you using? And why L1.5 instead of L2.0?

✓ We agree and modified the text as below. Unfortunately it was not possible to perform further statistical analysis (like we did for SYNOP) in validation against AERONET because the comparison is pixel-based and does not include a circular area around the station where we could estimate cloud fraction.

The reason for selecting level 1.5 instead of level 2.0 is that, level 1.5 data are cloud screened but level 2.0 data are quality assured. This means if we use level 1.5 data and check whether we have aerosol

measurements or not (having aerosol measurements means cloud-free condition) we could have information of cloudiness. But, in level 2.0 data, missing aerosol data could also be due to low quality of measurements.

- ✓ Modification: Line: 492-495: We also validated ASCIA cloud identification results with AERONET level 1.5 measurements. Because level 1.5 data are cloud screened. The procedure of this validation includes 2 steps: (1) covering AERONET observed AOT to a cloud flag (AOT is provided in AERONET only in cloud-free conditions); (2) Validation of ASCIA with AERONET cloud flag. In 86.1 % of 36 studied scenes over Svalbard, both ASCIA and AERONET confirm the presence of clouds.

Reviewer #2: Technical comments L151 The SLSTR revisit time is 1.9 day at the equator with one satellite and 0.9 day with two satellites, not single/dual view.

- ✓ Done.
- ✓ Modification: lines 161-162: This yields global revisit times of 1.9 days at the equator with two satellites and 0.9 day with one satellite.

Reviewer #2: Table 3. The title of the second column should be something like "Test"
- ✓ Done.
- ✓ Modification: line 698.

| Surface Type | Test | Description |
| --- | --- | --- |
| Water | $R_{0.87} < 11\%$ & NDSI$\geq$0.4 | MODIS snow and ice mapping ATBD (Hall et al., 2001) |
| Sea-ice | $R_{0.87} > 11\%$ & NDSI$\geq$0.4 | |
| Land | $R_{3.7}>0.04$ & $R_{0.66} < 0.2$ || NDSI<0.4 | Allen et al.,1999 |
| Snow | $R_{3.7}\leq0.04$ | Allen et al., 1999 |

Reviewer #2: Figures 1-2, fig. from 5 to 13 and fig. 16 should indeed be larger.
- ✓ Done.

Reviewer #2: L339-347 please simplify these lines. Throughout your manuscript sentences are often too long, but here they really affect the readability. Simplify the lines here and maybe add more information on the caption of the figures (for example the exact coordinates of the corners)
- ✓ Done.
- ✓ Modification: line 357-362: A representative example of the block level (25×25 km$^2$) and scene level (1×1 km$^2$) results of ASCIA applied to AATSR observations is shown in Fig. 5. This example was selected to show the performance of the algorithm in presence of different surface conditions: 1) one scene is over a combination of fairly homogeneous snow cover, land, ocean, sea-ice and cloud scene at north-west of

Greenland taken on the 18 May 2008; 2) another example is over a surface with highly variable topography over Svalbard with relatively higher solar zenith angle (>80°) on the 1 March 2008. As we discussed earlier, the ambiguity of the PCC analysis over homogeneous surfaces on the right and left side of the AATSR scene in middle panel of Fig. 5, is compensated in the right panel by using additional information from 3.7 μm channel.

✓ Line 707-711: Figure 5. Examples of the results of ASCIA on AATSR observations on the scenes over Greenland (upper panels) between (75°N, 48°W), (75°N, 75°W), (81°N, 48°W), (81°N, 75°W), taken on the 18 May 2008 and Svalbard (lower panels) within (75°N, 4°E), (75°N, 32°E), (81°N, 4°E), (81°N, 32°E) (lower panels), on the 1 March 2008, Left panels: RGB images, middle panels: Cloud detection at block level ($25 \times 25$ km$^2$), right panels: cloud detection at scene level.

**Reviewer #2: L16 reflection at 3.7 um**

✓ Done.

✓ Modification: line 16-17: Subsequently, the reflection at 3.7 $\mu$ m is used for accurate cloud identification at scene level at either $1 \times 1$ km$^2$ or $0.5 \times 0.5$ km$^2$.

**Reviewer #2: L18 e,g. e.g.**

✓ Done.

✓ Modification: line 17-18: The ASCIA data product has been validated by comparison with independent observations e.g. surface synoptic observations (SYNOP)

**Reviewer #2: L32 Though the attribution of the origins of this phenomenon**

✓ Done.

✓ Modification: line 32: Though the attribution of the origins of this phenomenon is controversially discussed.

**Reviewer #2: L76 the aim is**

✓ Done.

✓ Modification: line 79-80: … where the aim is to simultaneously retrieve aerosol and surface properties.

**Reviewer #2: L104 it is also planned to apply it to the observations acquired by SLSTR**

✓ Done.

✓ Modification: line 111-112: It is also planned to apply it to the observations acquired by SLSTR onboard Sentinel-3A and Sentinel-3B launched in 2016 and 2018 respectively which provide continuity of AATSR observations.

**Reviewer #2: L123 In the upper right**

✓ Done.

✓ Modification: line132-133: For example, in the upper right panel in Fig. 1 the large drop of reflectance over snow/ice created a notable contrast…

✓ Done.

✓ Modification: line 140-141: Each scene is imaged twice.

Reviewer #2: L142 These algorithms are typically not optimized

✓ Done. We deleted this line.

✓ Done.

✓ Modification: line 183-184: For example, they are almost absent in the central parts of the Arctic Circle as is shown in Fig. 3.

✓ Done.

✓ Modification: line 203-205: AERONET is a network of approximately 700 ground-based sun photometers established by National Aeronautics and Space Administration (NASA) and PHOtométrie pour le Traitement Opérationnel de Normalisation Satellitaire (PHOTONS).

✓ Done.

✓ Modification: line 222-223: To describe the computational procedure developed, we assume x, y to be two random variables, then the PCC can be written as…

✓ Done.

✓ Modification: line 223: …a function of the covariance of x and y which is normalized by square root of their variances.

✓ Modification: line 259.

✓ Done.

Reviewer #2: L254 Here you should compact everything in one sentence

✓ Done.

✓ Modification: line 273: where $R_{3.7}$ is the reflectance i.e. the ratio of scattered radiance to incident solar radiance; L measured radiance at 3.7 μm, $B_{3.7}(T_{11})$ the Planck function radiance (the contribution from

thermal emission at 3.7 μm) $T_{11}$ measurements at 11 μm; $F_{3.7,0}$ the solar constant at 3.7 μm and $\mu_0$ the cosine of solar zenith angle.

Reviewer #2: L285 AATSR provides

- ✓ Done.
- ✓ Modification: line 304: AATSR provides more data over higher latitudes, which increase in spring and summer time due to longer polar days and solar illumination.

Reviewer #2: L310 found that a PCC of 06

- ✓ Done.
- ✓ Modification: line 327-329: …we defined a lower threshold for PCC of 0.4 over the Arctic region and found that a PCC of 0.6 is appropriate for middle latitudes based on a number of statistical analyses.

Reviewer #2: L319 ASCIA starts looking for remaining small cloud scenes within a block, i.e. scenes . . . (R3.7 >0.04)

- ✓ Done.
- ✓ Modification: line 338: ASCIA starts looking for remaining small cloud scenes within a block, i.e. scenes with $R_{3.7}$ …

Reviewer #2: L333 it is important to note that one scene, . . .

- ✓ Done.
- ✓ Modification: line 351: It is important to note that one scene,…

Reviewer #2: L334 The latter mix with soil and becomes

- ✓ Done.
- ✓ Modification line 352-353: The latter mix with soil and becomes dark enough to be filtered out from the snow class. Sea-ice is distinguished from water on the basis of its greater brightness…

Reviewer #2: L339 a representative example

- ✓ Done.
- ✓ Modification: line 357: A representative example of the block level 25×25 $km^2$ and scene level 1×1 $km^2$ results of ASCIA…

Reviewer #2: L373 cloud free scene which ISTO failed to detect but are correctly labeled by ASCIA.

- ✓ Done.
- ✓ Modification: line 398-399: On the other hand, reddish scenes show cloud free case which ISTO fails to detect but are correctly labeled by ASCIA as cloud free.

Reviewer #2: L393 Both the ESA and ISTO

- ✓ Done.

✓ Modification line 416: Both the ESA and ISTO cloud products showed good results for this case with the exception of undetected thin cloud scenes which are falsely labeled as clear snow by ISTO.

Reviewer #2: L447 would be expected from SYNOP

✓ Done.

✓ Modification: line 486-487: As discussed earlier an error of ±1 to ±2 okta would be expected as the accepted accuracy range from SYNOP cloud cover values due to man-made nature of its observation and viewing conditions

**A cloud identification algorithm over the Arctic for use with AATSR/SLSTR measurements**

Soheila Jafariserajehlou[1], Linlu Mei[1], Marco Vountas[1], Vladimir Rozanov[1], John P. Burrows FRS[1], Rainer Hollmann[2]

[revised manuscript text omitted]
 applied to on AATSR observations is shown in Fig. 5. This example was selected to show the performance of ASCIA in presence of different surface conditions: 1) one scene is over a combination of fairly homogenous snow cover, land, ocean, sea-ice and cloud scene at north-west of Greenland taken on the 18 May 2008; 2) another example is over a surface with highly variable topography over Svalbard with relatively higher solar zenith angle (>80°) on the 1 March 2008. on the scenes within the region over northwest of Greenland in spring time enclosed in the coordinates for four corners (75°N, 48°W), (75°N, 75°W), (81°N, 48°W), (81°N, 75°W) taken on the 18 May 2008 are shown in Fig. 5. This example selected to show the performance of ASCIA over a combination of fairly homogenous snow cover, land, ocean, sea-ice and cloud. As we discussed earlier, the ambiguity of the PCC analysis over homogeneous surfaces on the right and left side of the AATSR scene in middle panel of Fig. 5, is entirely compensated in the right panel by using additional information from 3.7 μm channel. Another example over a surface with highly variable topography in March with relatively higher solar zenith angle (>80°) is selected over Svalbard enclosed in the coordinates for four corners (75°N, 4°E), (75°N, 32°E), (81°N, 4°E), (81°N, 32°E) taken on the 1 March 2008.

**5   Results and validation**

**5.1 The comparison to space-borne products**

In this study, we applied our recently developed ASCIA to identify cloud in the scenes using AATSR L1b (TOA reflectance) and SLSTR L1b gridded data. The input file to the process chain is one scene of AATSR L1b product the output comprises 5 classes of surface types including snow/ice, sea ice, water, cloud and land. The procedure of

surface classification is explained in Sect. 4. The location and time of selected case studies are used to show that the identification of cloud by our new ASCIA is adequate. In this regard, the AATSR data are selected from several years starting from 2006, during strong Arctic haze episode, which originated predominantly from agricultural fires burning in Eastern Europe. The event has been reported in previously (Law et al., 2007). A second episode in 2008 is also considered for which validation data are available from SYNOP stations. Three months of data from  March, May, and July  have been acquired over Greenland and Svalbard to assess the performance of  ASCIA in a wide range of solar zenith angles (60°-85°), surface and atmospheric conditions observed at high latitudes. In order to  various surface types in the Arctic take into account, we selected case studies including, highly variable topography and fairly homogeneous snow cover, coast lines, land and ocean along snow and ice covered surface. The designed criteria for  ASCIA are optimized for  various regions over the Arctic observed under different solar illumination conditions. Polar night and transition seasons at low light conditions are excluded from our retrievals.  
[revised manuscript text omitted]
. To define the optimal maximum temporal difference between SYNOP and satellite data, other comparable validation activities used different temporal intervals like 10 min (Werkmeister et al., 2015), 15 min (Musial et al., 2014), 1 h (Dybbroe et al., 2005) and 4 h (Meerkötter et al., 2004). The investigation and results in the previous publications indicate that temporal difference in validation of satellite retrievals against SYNOP may vary based on meteorological conditions. Allowing only a small temporal difference between measurement datasets (here: SYNOP and ASCIA) ensures as close as possible temporal resemblance but can introduce a significant sampling error due to limited number of validation cases (Bojanowski et al., 2014). According to Bojanowski et al. (2014) a temporal difference of 90 min to compare with SYNOP measurements at temporal resolution of 3 h minimizes the sampling error (Bojanowski et al., 2014). However, potential longer temporal difference will introduce an error which should be considered along other sources of uncertainty (different viewing perspective, different spatial footprint and etc.). In this study, the maximum allowed temporal difference between the ASCIA retrievals and SYNOP measurements is small being below ±20 minutes in most cases and generally does not exceed ± 45 minutes. The difference in the time of satellite and SYNOP measurements is small being below ±20 minutes in most cases and generally does not exceed ± 45 minutes. To compare surface measurement from SYNOP hemispheric view with the cloud identification at a spatial resolution $1\times1$ km$^2$ resolution satellite measurement, we calculated cloudiness as the percentage of cloudy scenes within a window of $20\times20$ km$^2$ around each SYNOP station. This is a similar distance to that used in previous studies to validate satellite based cloud identification SYNOP or similar surface measurements (Kotarba, 2017; Werkmeister et al., 2015; Minnis et al., 2003). The cloud detection data product was then compared to the three months (March, May and July) of SYNOP observations. These comprise 100 measurements over Svalbard and Greenland.

[revised manuscript text omitted]

| | Criteria | |
|---|---|---|
| Cloud data | within ±2 oktas | within ±1 okta |
| ASCIA vs. SYNOP | 96 % agreement
4 % disagreement | 83 %
agreement
17 %
disagreement |
| ESA vs. SYNOP | 68 % agreement
32 % disagreement | 50 %
agreement
50 %
disagreement |

[Figure]

[Figure]

701 **Figure 1.** Upper left: the false colour RGB image of AATSR (using 0.67, 0.87 and 0.55 μm channels) over Svalbard, 10 May
702 2006, upper right: 1.6 μm reflectance, lower left: 3.7 μm reflectance, lower right: 11 μm brightness temperature.

[Figure]

[Figure]

Figure 2. Upper left: the RGB false colour image (using 0.67, 0.87 and 0.55 µm channels) of SLSTR over Svalbard, 18 April 2017, upper right: 1.37 µm reflectance, lower left: 1.6 µm reflectance, lower right: 3.7 µm reflectance.

[Figure]

705 **Figure 3.** SYNOP network coverage over the Arctic, the dark blue points indicate the location of SYNOP stations.

[Figure]

**Figure 4.** The schematic flowchart of ASCIA.

[Figure]

**Figure 5.** Examples of the results of ASCIA on AATSR observations on the scenes over Greenland (upper panels) between (75°N, 48°W), (75°N, 75°W), (81°N, 48°W), (81°N, 75°W), taken on the 18 May 2008 and Svalbard (lower panels), within (75°N, 4°E), (75°N, 32°E), (81°N, 4°E), (81°N, 32°E) (Lowe panels), taken on the 18 May 2008 andon the 1 March 2008 respectively, Left panels: RGB false colour images (using 0.67, 0.87 and 0.55 μm channels), middle panels: Cloud detection at block level ($25\times25$ km$^2$), right panels: cloud detection at scene level.

[Figure]

Figure 6. (a) The RGB false colour image (using 0.67, 0.87 and 0.55 μm channels) of AATSR over northern Greenland, 24 May 2008, (b) Nadir cloud flag from AATSR L2 product, (c) cloud detection based on spectral shape of clear snow, (d) cloud detection of ASCIA, (e) difference between ISTO and ASCIA

[Figure]

Figure 7. (a) The RGB false colour image (using 0.67, 0.87 and 0.55 μm channels) of AATSR over Svalbard, 10 May 2006, (b) Nadir cloud flag from AATSR L2 product, (c) cloud detection based on spectral shape of clear snow, (d) cloud detection of ASCIA, (e) difference between ISTO and ASCIA.

[Figure]

[Figure]

**Figure 8.** (a) The RGB false colour image (using 0.67, 0.87 and 0.55 μm channels) of AATSR over Svalbard, 18 March 2008, (b) Nadir cloud flag from AATSR L2 product, (c) cloud detection based on spectral shape of clear snow, (d) cloud detection of ASCIA, (e) difference between ISTO and ASCIA.

[Figure]

**Figure 9.** (a) The RGB false colour image (using 0.67, 0.87 and 0.55 μm channels) of AATSR over Svalbard, 6 July 2008, (b) Nadir cloud flag from AATSR L2 product, (c) cloud detection based on spectral shape of clear snow, (d) cloud detection of ASCIA, (e) difference between ISTO and ASCIA.

724

[Figure]

**Figure 10.** (a) The RGB false colour image (using 0.67, 0.87 and 0.55 μm channels) of AATSR over Svalbard, 3 May 2006, (b) Nadir cloud flag from AATSR L2 product, (c) cloud detection based on spectral shape of clear snow, (d) cloud detection of ASCIA, (e) difference between ISTO and ASCIA.

[Figure]

**Figure 11.** Left panel: RGB false colour image (using 0.67, 0.87 and 0.55 μm channels) of AATSR over Svalbard, 14 July 2008, 16h 40min 45s, middle panel MODIS cloud mask algorithm retrieved data: 1- cloudy, 2- probably cloudy, 3- probably clear, 4- clear, (red rectangle shows the coverage of AATSR) for 16h 25 min, right panel: the results for the cloud detection of ASCIA.

[Figure]

**Figure 12.** Left panel: RGB false colour image (using 0.67, 0.87 and 0.55 µm channels) of AATSR over Greenland, 18 May 2008, 23h 13min 38s, middle panel: MODIS cloud mask: 1- cloudy, 2- probably cloudy, 3- probably clear, 4- clear, (red rectangle shows the coverage of AATSR) for 23h 5min, right panel: Cloud detection of ASCIA.

[Figure]

**Figure 13.** Left panel: The RGB false colour image (using 0.67, 0.87 and 0.55 μm channels) of SLSTR over Svalbard, 18 April 2017, 10hr 15min 6s, Middle panel: MODIS cloud mask: 1- cloudy, 2- probably cloudy, 3- probably clear, 4- clear, (red rectangle shows the coverage of AATSR) for 11h 30m, right panel: Cloud detection of ASCIA.

[Figure]

738     **Figure 14.** Density plot of occurrences of the CFC by ASCIA as a function of SYNOP.

[Figure]

739     **Figure 15.** Histogram of CFC differences (blue: ASCIA minus SYNOP; red: ESA cloud product minus SYNOP).

[Figure]

740 **Figure 16.** CFC in percent by ASCIA (red), SYNOP (blue) and ESA Cloud Product (green) for 100 scenarios of March, May and
741 July 2008 over Svalbard and Greenland. Light blue error bars show the range of percentage values for each okta from SYNOP
742 measurements.